# Nanosecond optical switching and control system for data center networks

Xuwei Xue [ID][1✉] & Nicola Calabretta[1]

Electrical switching based data center networks have an intrinsic bandwidth bottleneck and, require inefficient and power-consuming multi-tier switching layers to cope with the rapid growing traffic in data centers. With the benefits of ultra-large bandwidth, high-efficient cost and power consumption, switching traffic in the optical domain has been investigated to replace the electrical switches inside data center networks. However, the deployment of nanosecond optical switches remains a challenge due to the lack of corresponding nanosecond switch control, the lack of optical buffers for packet contention, and the requirement of nanosecond clock and data recovery. In this work, a nanosecond optical switching and control system has been experimentally demonstrated to enable an optically switched data center network with 43.4 nanosecond switching and control capability and with packet contention resolution as well as 3.1 nanosecond clock and data recovery.

[1] Electro-Optical Communications group, IPI research institute, Eindhoven University of Technology, 5612 AZ Eindhoven, Netherlands. ✉email: x.xue.1@tue.nl

The escalation of traffic-boosting applications and the scale out of powerful servers have significantly increased the traffic volume inside the data centers (DCs)[1]. Consequently, each aggregation switching node in the data center network (DCN) has to handle multiple Tb/s to hundreds of Tb/s traffic. Along with the increasing demands of higher switching bandwidth, emerging latency-sensitive applications are also imposing stringent requirements of low latency[2] to DCNs. However, due to the limited maximum bandwidth per pin of the CMOS chips, as well as the limited chip number that can be used on a single package in electrical switching technologies, it is hard to linearly increase bandwidth[3]. New technologies, such as Silicon Photonics[4], 2.5D/3D packaging[5], and co-packaging[6], are being investigated to scale the I/O bandwidth. However, before these technologies become viable, a number of challenges have to be solved, e.g., the high complexity to package external laser sources and fiber coupling, and the high manufacturing (including both packaging and testing) costs.

As a counterpart, switching the traffic in the optical domain has been investigated considerably as a solution to overcome the bandwidth bottleneck and latency issues in DCNs[7,8]. Benefiting from the optical transparency, the optical switching with high bandwidth is independent of the bit rate and data format of the traffic. Wavelength division multiplexing (WDM) technology can be employed to boost the optical network capacity at a superior power-per-unit bandwidth performance[9]. In addition, the optical switching networks eliminate the dedicated interfaces for modulation-dependent processing, achieving fast and high processing efficiency[10]. Furthermore, eliminating the power-consuming optical-electrical-optical conversions at the switch nodes significantly improves the energy and cost-efficiency[11]. All these benefits can be exploited to flatten the DCN topology and overcome the hierarchical architecture with associated large latency and low throughput[12].

Multiple optical switching techniques have been proposed and investigated, of which the micro-electro-mechanical (MEMS)-based slow switches are seeing penetration into data centers, providing reconfigurable high-bandwidth and data-rate transparency channels[13]. However, the tens of milliseconds of switching time have strictly confined the applications to well-scheduled and long-lived tasks. Considering the time-varying traffic bursts and high fan-in/out hotspots patterns in DCNs, the slow optical switches providing static-like and pairwise interconnections would only be beneficial as supplementary switching elements. Contrarily, fast optical switches with nanosecond switching time in support of the packet-level operations, such as semiconductor optical amplifier (SOA)-based optical switch, can be exploited to realize DCNs with high network throughput, flexible connectivity, and on-demand resource utilization[14,15].

Despite the promises held by the fast optical switching technologies, the practical implementation of the nanosecond optical switching DCNs is actually facing several challenges. As the main unresolved challenge, a nanosecond scale control system is essential for the switch control and fast forwarding of the data traffic[16]. Fast switch reconfiguration time including both hardware switching time (on the order of nanoseconds for SOA-based switch) and controlling overhead time is essential as it determines the network throughput and latency performance. Moreover, the reconfiguration time should be independent of the DCN scale. This requires fast processing of the optical labels that carry the destinations of the packets at the switch controller on the order of nanoseconds, which subsequently reconfigures the optical switch for the data packets forwarding. However, the synchronization requirement between the processed optical labels at the switch controller and the delivered optical packets at the optical switch strongly limits the implementation of the nanosecond

switching control[17]. To complete the switch control, packet contention resolution is another unsolved critical challenge that needs to be addressed[18]. The electrical switches employ random access memories (RAM) to buffer the conflicted packets and then solve contentions. Because no effective RAM exists in the optical domain, the conflicted packets at the optical switch will be dropped and thereby cause high packet loss. Despite several approaches that have been proposed to overcome such issues, based either on optical fiber delay lines (FDLs)[19] or deflection routing[20], none of them is practical for large-scale DCN implementation.

Unlike the point-to-point synchronized connections between any paired electrical switches, optical switches create only momentary physical links between sources and destination nodes[17]. Therefore, in a packet-based optical switching network, the clock frequency and phase of the data signal vary packet by packet. Thus, nanosecond burst-mode clock and data recovery (BCDR) receivers are demanded to recover the clock frequency and phase on the packet base. The BCDR locking time determines the number of preamble bits that can dramatically reduce the network throughput, especially for the intra-DC scenario where many applications produce short traffic packets[21]. BCDR receivers have been extensively studied in the context of Passive Optical Networks and architectures based on gated oscillators or over-sampling have been shown to achieve nanoseconds locking time[22]. These techniques, however, increase the complexity and cost of the transceiver design and need to be re-evaluated for higher data rates, although initial results are encouraging[23]. The burst-mode links proposed in ref. [24] can be dynamically reconfigured to support the fast optical switching. However, the control plane of this solution does not resolve the packet contention, which could cause high packet loss.

The above mentioned challenges in terms of nanosecond switching control, packet contention resolution, and fast CDR locking have been the roadblock to the deployment of fast optical switches in the DCNs. In this work, we propose and experimentally demonstrate a nanosecond optical switching and control system to comprehensively solve all these issues that prevented deployment of nanosecond optical DCN with packet-based operation. The nanosecond optical switching and control system is based on a combination of a label control and synchronization mechanism for nanosecond switch control, an Optical Flow Control protocol to resolve packets contention, and a precise clock distribution method for nanosecond data packet recovery without the deployment of BCDR receivers. Experimental results validate that the label control system is capable to distribute the clock frequency from the switch controller to all the connected top of racks (ToRs) allowing 3.1 ns data recovery time with no BCDR receivers, and 43.4 ns overall switching and control time of the DCN.

## Results

**Label control mechanism.** The proposed nanosecond optical switching and control system is schematically illustrated in Fig. 1a. At each rack, the Ethernet frames generated by the H servers are first processed by the FPGA-based Ethernet switch. Based on the Ethernet medium access control (MAC) address, frames destined to servers in the same rack (intra-rack traffic) are directly forwarded to the intra-rack servers. While frames destined to servers in different racks (inter-rack traffic) are stored in the electrical RAM at each network interface controller (NIC). The copies of the stored frames are sent to the data packet processor which aggregates the Ethernet frames with the same destination to generate an optical data packet. Meanwhile, a corresponding optical label is generated at the label packet

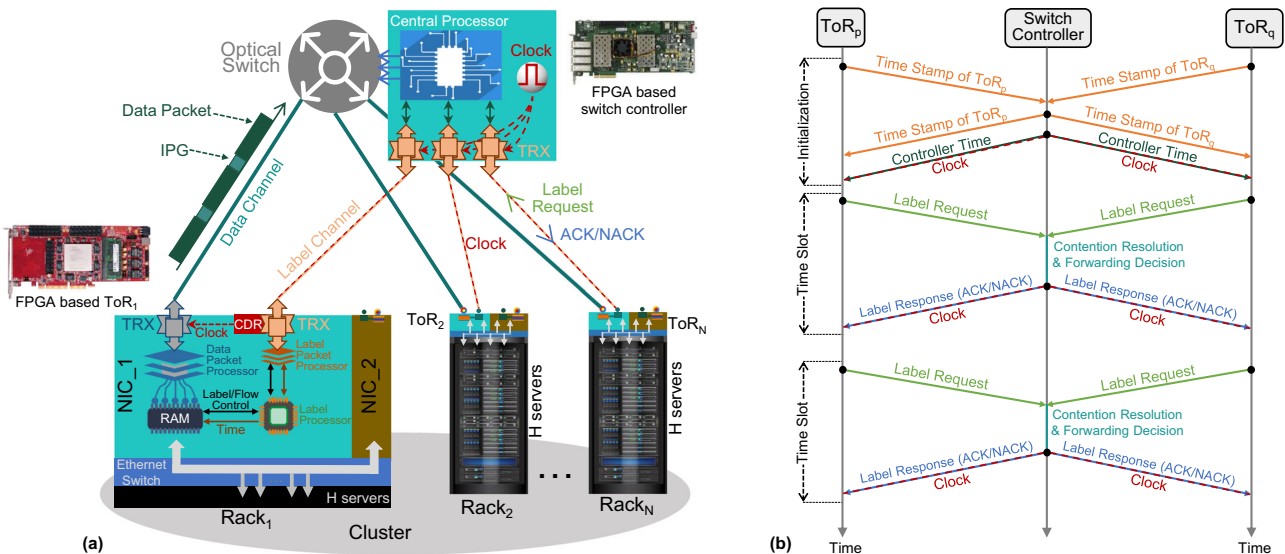

**Fig. 1 Concept of the proposed optical switching and control system. a** Nanosecond optical switching and control system exploiting label control mechanism, Optical Flow Control protocol and clock distribution mechanism. In the proposed system, N racks are grouped into one cluster and each rack hosts H servers. Each top of rack (ToR) consists of one Ethernet switch and two network interface controllers (NICs). The field programming gate array (FPGA)-based ToR₁ and FPGA-based switch controller are zoomed to show the detailed functional components. Data channels carrying the optical packets from the ToRs are cross-connected by the optical switch, and the label channels are used to connect the ToRs to the switch controllers. IPG inter-packet gap, ACK acknowledgment, NACK negative acknowledgment. **b** Signals exchange between switch controller and ToRs. At system initialization state, the timestamps are exchanged between the switch controller and ToRs to automatically measure the fiber delay and distribute the time of switch controller to connected ToRs to implement the synchronous slotted mechanism. Afterwards, the label request and response packets are exchanged to implement the label control mechanism and Optical Flow Control protocol at each time slot. The clock is also continuously distributed by the label response signals to the ToRs to synchronize the network clock. The label control mechanism, Optical Flow Control protocol and clock distribution mechanism are employed on the label channels with less resource occupied.

processor based on the ToR look-up table to indicate the destination information and forwarding priority of the optical data packet. The optical label (label request packet) is delivered to the switch controller via the label channel. After a small time offset, the corresponding optical data packet is sent out to the optical switch via the data channel. Based on the received label requests, the switch controller resolves the packet contention, and then generates the control signals to reconfigure the fast optical switch. Note that the label requests also carry information of the packet priority, in case the various quality of service should be provided. Benefiting from the parallel processing capability of FPGA, the label requests can be processed within a few clock cycles (nanoseconds) to implement the fast switching control. The optical data packets are then forwarded by the optical switch to the ToR destinations.

**Time synchronization**. In the time-slotted optical packet switching network, the slotted optical packets generated by the ToRs have to arrive aligned at the optical switches. This requires precisely unifying the time for all the ToRs. During the system initialization state as shown in Fig. 1b, first each ToR sends the timestamp with initial time ($T_{L1}$) to the switch controller via the label channels. The timestamps are processed at the controller and sent back to the source ToRs. The corresponding ToR records the time ($T_{L2}$) when the timestamp is received. Based on the time offset ($T_{offset} = T_{L2} - T_{L1}$), and the processing delay ($T_{processing}$) inside the FPGA-based ToR and switch controller, the physical fiber transmission delay ($T_{fiber}$) of the label channel can be automatically measured as $T_{fiber} = (T_{offset} - T_{processing})/2$. Thus, the switch controller sends the controller time ($T_{controller}$) to all the connected ToRs. Once the controller time is received at each ToR, the ToR time of ($T_{ToR}$) will be updated by compensating the received controller time with the measured fiber delay

and the FPGA processing time ($T_{ToR} = T_{controller} + T_{fiber} + T_{processing}/2$). This mechanism guarantees all the ToRs with identical time reference inherited from the switch controller. Therefore, the optical labels and optical data packets can be sent out aligned with the timeline, guaranteeing the synchronization of the optical labels at the switch controller and the data packets at the optical switch.

**Optical flow control (OFC) protocol**. Considering the lack of optical buffer at the optical switches, an Optical Flow Control (OFC) protocol is developed for resolving the packet contentions when multiple optical data packets have the same destination. Once contentions occur, the data packets with higher priority will be forwarded to the destination ToRs while the conflicted packets with lower priority will be forwarded to the ToRs with no destination requesting. This kind of packet forwarding mechanism guarantees the receivers at each ToRs to receive a continuous traffic flow at every time slot. The developed OFC protocol is deployed on the bidirectional label channels between the switch controller and the ToRs. After the contention resolution, the switch controller sends back to each ToR an ACK signal (indicating packet successfully forwarded by the optical switch) or NACK signal (packet forwarded to the un-destined ToRs). Based on the received ACK/NACK signal, the ToR label packet processor will release the stored data packet from the RAM (ACK signal) or trigger the data packet processor to retransmit the optical packet (NACK signal). Moreover, to prevent the packet loss at the overflowed buffer, the layer-2 flow control mechanism, like the Ethernet PAUSE frame and priority-based flow control (PFC), can be integrated in this control system. The ToR can monitor the buffer occupation ratio of each buffer block in real-time[25]. When the monitored buffer is over the predefined queuing threshold, the ToR can generate a PAUSE or PFC frame

and send it back to the corresponding source on the reverse path (label channel) with the normal Ethernet frames to pause the traffic transmitting.

**Clock frequency distribution.** In this proposed approach, each of the bidirectional label channels is a continuous link not only used to send the label requests from ToRs to the switch controller and the ACK/NACK signals from the switch controller to the ToRs, but it is also used for clock distribution from the switch controller to the ToRs to synchronize the system clock frequency. As shown in Fig. 1a, the onboard clock source of the switch controller is employed as a master clock to be distributed to the ToRs via the bidirectional label channels. At each NIC of the ToRs, the clock is recovered from the continuous ACK/NACK streaming by a conventional CDR receiver. The continuous clock distribution in the proposed system is implemented on the independent label channels to each ToR. The distributed clock is recovered and cleaned at the CDR block (equipping a PLL inside) even passing the fiber with different lengths on label channels, as shown in Fig. 1a. The recovered and cleaned clock without jitter is then used to drive the transceivers of the data channels. Therefore, the quality of the continuous distributed clock (jitter and cabling) is not affected by the counts of the label channels (the scalability of the system). In this way, the clock with the same frequency is distributed and used in all the ToRs not only to transmit the optical packets and the optical labels, but also to implement the nanosecond recovery of the optical data packets. Indeed, once all the network ToRs have the clock with the same frequency, the receiver only needs to align the clock phase of the incoming data, which can be achieved within a few tens of bits (few nanoseconds) preventing the need for a time-consuming clock frequency recovery.

It should be noted that the CDR circuits at the conventional receivers need to receive continuous data traffic to maintain the recovered clock with good quality. To guarantee this, the switch controller, which has the full vision of the traffic from the ToRs, exploits the multicast capability of the optical switch to forward the conflicted packet with lower priority to one un-destined ToR

to fill the empty slot. Moreover, the system only needs to distribute the clock frequency to the ToRs in the sub-network, there is no requirement to align the clock phase as the CDR is done at each ToR in less than one clock cycle. This is an advantage with respect to other techniques[26], where the phase alignment requires more complex network interconnections and extra devices to guarantee the precise positional relationship between the TX and RX sides, limiting the practical implementation in large scale DCNs. Furthermore, the clock carried by the label channel does not need to be distributed at full line rate, because the clock can be frequency multiplicated at the CDR block according to the transceivers requirements of the data channels.

**Network structure.** The system shown in Fig. 1a considers only the single cluster operation. Large-scale multi-cluster optical DCNs can also be built based on this proposed nanosecond optical switching and control system as structurally shown in Fig. 2, where inter-cluster optical switch (ES) is employed to connect ToRs locating in different clusters. Ethernet frames destined to servers located in the same cluster (intra-cluster) or different cluster (inter-cluster) are aggregated into optical data packets at intra-NICs or inter-NICs, respectively. Single-hop direct interconnection is provided by the intra-cluster switches (IS) for the intra-cluster communication. For the inter-cluster traffic, two-hops interconnection is sufficient for the communication completion. As an instance shown in Fig. 2, traffic from $ToR_1$ in $cluster_1$ destined to $ToR_M$ in $cluster_2$ could be first forwarded by the inter-cluster switch $ES_1$ to the intermediate $ToR_{N+1}$ in cluster 2. The $ToR_{N+1}$ Ethernet switch, based on the destination address, forwards the data packets to the intra-NIC so that the packets are delivered to the $ToR_M$ via the intra-cluster switch $IS_2$. Another two-hops link for this inter-cluster communication is $Cluster_1ToR_1 \leftrightarrow IS_1 \leftrightarrow Cluster_1ToR_M \leftrightarrow ES_M \leftrightarrow Cluster_2ToR_M$. Moreover, the Ethernet switch at each ToR monitors the traffic volume of intra-cluster and inter-cluster communications by reading the destination MAC address[27], and then accordingly assigns the adaptable optical bandwidth to the intra-cluster and inter-cluster links.

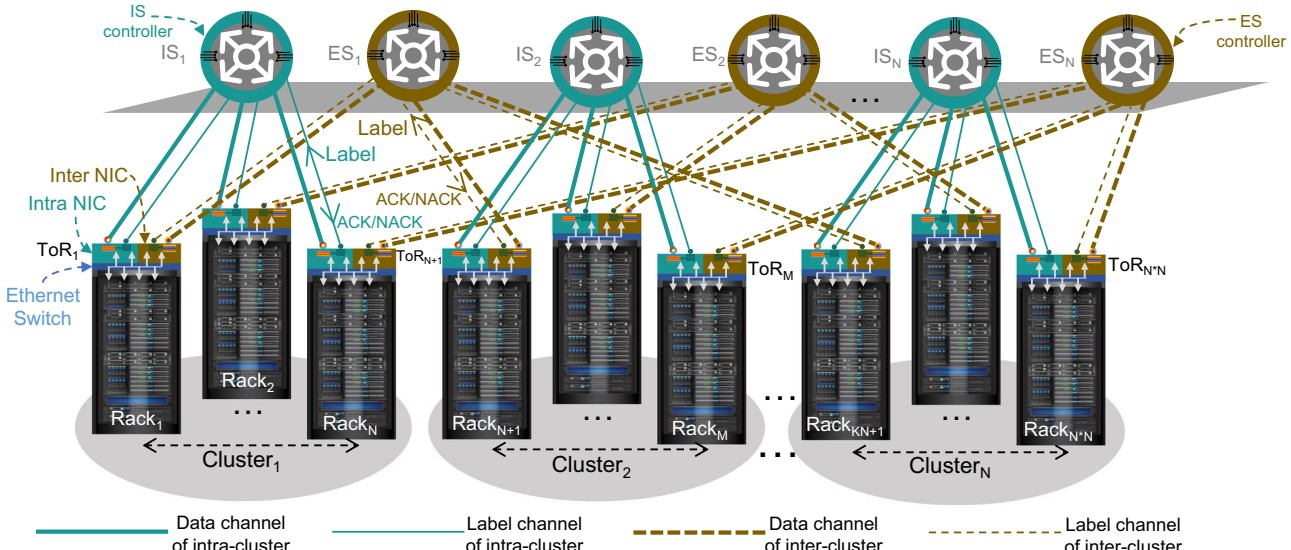

**Fig. 2 Large-scale optical DCN based on the proposed optical switching and control system.** The total N × N racks are divided into N clusters in the large-scale optical DCN. The two NICs at each ToR can be divided into two categories: NIC of intra cluster (intra-NIC) and NIC of inter cluster (inter-NIC). The N N × N intra-cluster optical switches (IS) connecting the intra-NICs in the same cluster are dedicated for the intra-cluster traffic communication. The inter-NIC of the i-th ToR in different cluster are interconnected by the i-th (1 ≤ i ≤ N) inter-cluster optical switch (ES), which are responsible for the forwarding of inter-cluster traffic.

Note that the intra-cluster interconnect network (consisting of the intra-NICs and intra-cluster switch) and the inter-cluster interconnect network (consisting of the inter-NICs and inter-cluster switch) are two independent sub-networks as shown in Fig. 2 by the solid and broken links, respectively. Each sub-network has an independent optical switching and control system with its own label control mechanism, OFC protocol and clock frequency distribution. This is important as the scalability of the optical switching and control system is per cluster scale and not for the whole DCN. This makes the proposed techniques fully distributed and scalable even for DCN with a very large number of ToRs.

**Experimental demonstration**. The experimental setup illustrated in Fig. 3a is built to assess and validate the performance of the nanosecond optical switching and control system. The network analyzer (Spirent) generates the Ethernet frames with variable and controllable traffic load, and the size of Ethernet frames are generated randomly between 64 bytes and 1518 bytes with an average size of 792 bytes. The fiber length between ToRs and switch nodes are different to validate the synchronized slot mechanism and the robustness of the switching system. The format of the optical packet with 2600 bytes in length is shown as the inset in Fig. 3b. The inter-packet gap (IPG) and the idle part of the payload are filled with pulse transition sequence ("1010…1010") to maintain the continuous stream of data, similar as in the Ethernet protocol. Note that the length of the optical data packets can be flexibly reconfigured to adapt to the variable workload, benefitting from the programmable capability of the FPGA-based hardware.

**Fast switching and control**. We first experimentally validate the initialization of the system with the time allocation, and then validate the nanosecond operation of the label controlling mechanism and the OFC protocol. At the system initialization state, the timestamps used to measure the fiber delay and the allocated time used to unify the ToR time are monitored at the switch controller as illustrated Fig. 3c. Once the ToRs have the unified time slot synchronization, ToRs transmit the label request signals that indicate the destination and forwarding priority of the associated data packets to the switch controller at every time slot. In this experimental case, the priority order of data packets is set as '1 > 2 > 3 > 4' where packets with order '1' have the highest priority. The FPGA-based switch controller resolves the contention according to the label requests and then sends the ACK (LabelResponse = LabelRequest) and NACK (LabelResponse ≠ LabelRequest) signal to the corresponding ToRs, respectively. The monitored traces in Fig. 3d show that the label request/response signals and the optical data packets are synchronized at the switch controller and optical switch, respectively. This validates the implementation of the synchronous slotted mechanism based on the accurate time allocation. As the packet contention instance shown in Fig. 3d, the label requests for $ToR_1$ and $ToR_2$ is 3 indicate that the data packets from $ToR_1$ and $ToR_2$ are destined to $ToR_3$ in time slot N. Given the higher priority, $ToR_1$ packet is forwarded to the $ToR_3$, while $ToR_2$ packet with lower priority is sent to $ToR_2$ to maintain the continuous stream traffic (this receives the packet at $ToR_2$ will be dropped once verified that its destination is $ToR_3$). Afterwards, $ToR_1$ receives an ACK signal (see label response in Fig. 3d) to release the stored packet in the electrical buffer, while $ToR_2$ receives the NACK signal to trigger the packet retransmission. In the next time slot N + 1, $ToR_1$

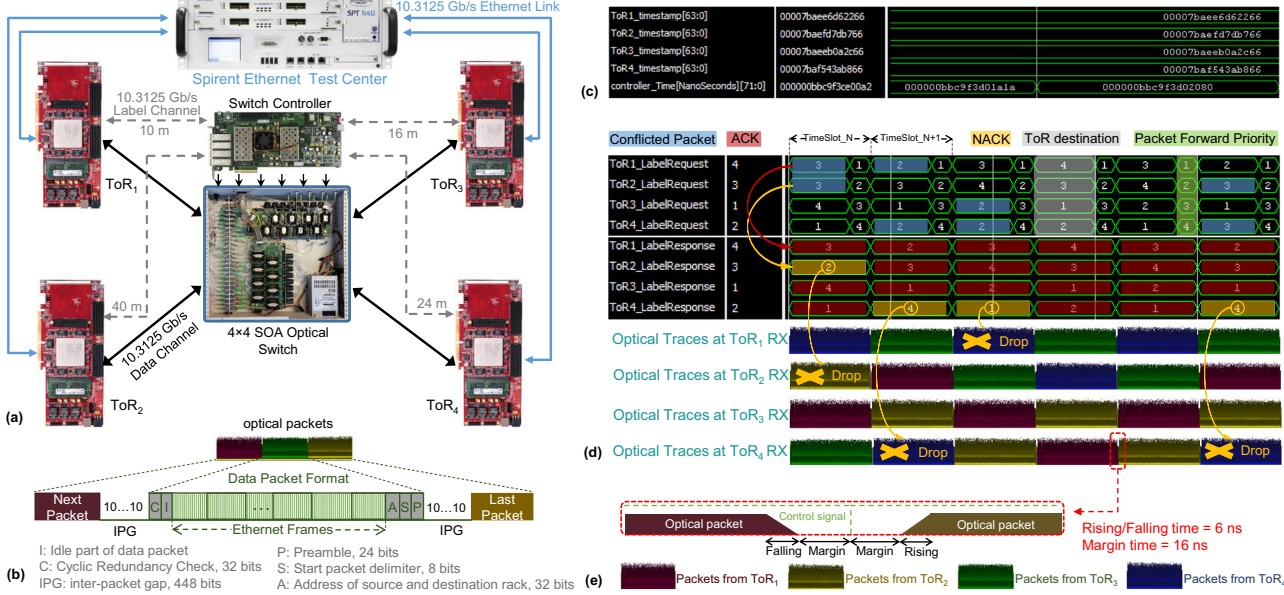

**Fig. 3 Experimental demonstration of the nanosecond optical switching and control system. a** Proof-of-concept experiment. The experimental demonstration consists of 4 FPGA-based (Xilinx VU095) ToRs implementing the functions of Ethernet Switch, intra-NIC and inter-NIC. The Ethernet interfaces at each ToR interconnect the Spirent Ethernet Test Center via 10.3125 Gb/s Ethernet links. One semiconductor optical amplifier (SOA)-based 4 × 4 optical switch with corresponding FPGA-based (Xilinx VC709) controller are utilized to interconnect all the ToRs. The data packets are delivered via the 10.3125 Gb/s data channels to the optical switch, which in turn forwards the packets to the destined ToRs configured by the FPGA switch controller. The 10.3125 Gb/s label channels efficiently implement the label controlling mechanism, OFC protocol and clock frequency distribution. **b**. Format of optical data packet. The packet consists of a 24 bits preamble, 8 bits start packet delimiter, 32 bits ToR source/destination address, 32 bits Cyclic Redundancy Check (CRC) sequence and the rest is the payload (aggregated Ethernet frames). The 32 bits ToR source/destination address is embedded for packet identification. The Cyclic Redundancy Check (CRC) is a 32-bit checksum calculated to provide error detection in the case of packet transmission collisions or link errors that could corrupt the data packet. **c** Monitored timestamps at switch controller. **d** Label signals monitored at switch controller and optical traces at receiver side of ToRs. **e** inter-packet gap time.

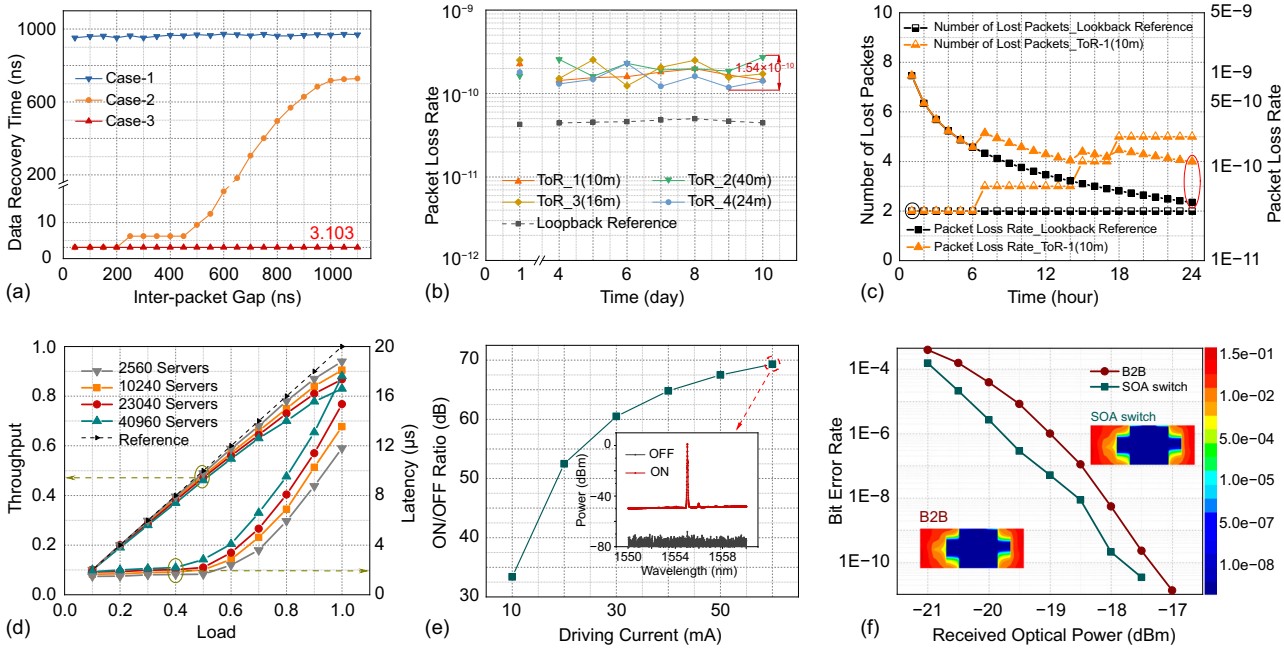

**Fig. 4 Network performance of the proposed optical switching and control system. a** Data recovery time as the function of the inter-packet gap length. Case-1: without clock distribution mechanism and without pulse transitions insertion; Case-2: With clock distribution mechanism and without pulse transitions insertion; Case-3: With clock distribution mechanism and with pulse transitions insertion. **b** 7 day packet loss rate measured on the optical link. The number of sent data packets to specific ToRs is recorded every day and the received packets at each specific ToR are counted as well. E.g., the total number of data packets delivered from $ToR_2$, $ToR_3$ and $ToR_4$ to $ToR_1$ is CT, and the count of correct packets after CRC checking at the destined $ToR_1$ is CR. Thus, the packet loss rate on the optical link to $ToR_1$ is calculated as (CT-CR)/CT. **c** The detailed packet loss rate on the 4th day. **d** Throughput and server-to-server latency for large-scale network scale. The label control mechanism, OFC protocol, and clock distribution are deployed in the OMNeT++ model, completely following the technical design. In this model, 40 servers are grouped in each rack and 6 WDM transceivers are deployed at each ToR, where each transceiver equips with 25 KB electrical buffer. **e** ON/OFF power ratio under different driving currents. **f** BER curve and eye diagram of the deployed SOA switch. The Xilinx IBERT IP Core is deployed in each FPGA-based ToR to measure the BER performance of the SOA switch. SOA semiconductor optical amplifier, B2B back-to-back.

sends out a new label request 2 while $ToR_2$ sends again the label request 3 until receiving the ACK signal. As schematically shown in Fig. 3e, the overall switching and control system is implemented in 43.4 ns, which consists of 12.4 ns label processing time, 3 ns switch driver delay, 6 ns switch rising time, 6 ns switch falling time, and 16 ns margin time of the switch control signal with respect to the optical packet.

**Fast clock and data recovery**. Next, the ultra-fast data recovery is investigated to validate the clock distribution mechanism and the filling empty slot protocol with pulse transitions insertion. Figure 4a shows the data recovery time as function of the IPG length. As a comparison, the data recovery time of the system without clock frequency distribution is measured as the reference. For this case, around 970 ns, mostly spent on the clock frequency recovery, are required to recover the correct data. By deploying the proposed clock distribution mechanism, as observed from Fig. 4a, the data can be correctly recovered within 6.206 ns for the scenario of IPG less than 450 ns. However, the CDR circuit will start losing the lock when the IPG without inserting pulse transitions is longer than 500 ns. Employing the clock frequency distribution mechanism and inserting the pulse transitions to fill empty IPG period, the results show a constant 3.1 ns data recovery time regardless of the IPG length, and without deploying the dedicated BCDR receivers.

**System stability**. The stability assessment of the clock distribution, synchronization of the slotted mechanism, and the effectiveness of the OFC protocol are investigated by counting the

packet loss on the optical links for 10 days at the traffic load of 0.8. The packet loss on the loopback link from $ToR_1$-TX to $ToR_1$-RX without passing the optical switch is also measured as the reference. Figure 4b validates that the packet loss for all the ToRs is less than 3.0E−10 after 10 days running. In addition, the packet loss for all the nodes maintains stability and the maximum fluctuation is less than 1.54E−10. The low and stable packet loss validates the accurate operations of the nanosecond switching and control system based on the OFC protocol, the slotted mechanism, and the robustness of the clock distribution, even for different fiber lengths are deployed between ToRs. Figure 4c shows the detailed packet loss rate of the 4th day for the loopback ($ToR_1$-TX to $ToR_1$-RX) and $ToR_1$ ($ToR_2$-TX, $ToR_3$-TX, $ToR_4$-TX to $ToR_1$-RX), respectively. Both the loopback and $ToR_1$ channel loss 2 packets at the initialization state while the $ToR_1$ losses 5 packets in total. The temporary time disorder at the initial state could introduce a couple of packet losses.

**Networking scalability**. To verify the networking capability of the proposed system, OMNeT++ simulation model of large-scale optical DCN is built based on the principles illustrated in Fig. 2. The experimentally measured parameters are used to program this simulation model to emulate the practical network operating scenario. The average link throughput and server-to-server latency performance as a function of the network scalability has been numerically investigated, as shown in Fig. 4d, where the throughput baseline is used as a reference. With respect to the baseline, the throughput of the network with 40,960 servers decreases 10% at the load of 0.7, because the intermediate ToRs in

the large-scale network have to forward the inter-cluster traffic from other ToRs, thus occupying the bandwidth for its own traffic. The server-to-server latency is below 3 μs at a load of 0.5 for the large scale (40,960 servers) network, and the numerical results validate a limited 9.46% latency degradation as the network scales from 2560 to 40,960 servers.

**SOA-based optical switch**. The N×N semiconductor optical amplifier (SOA) based optical switch is a broadcast and select (B&S) style. For each output port, only one SOA gate will be ON state when forwarding optical packets to this specific port. Other SOA gates at OFF state may not completely block the optical signal and then introduce the cross-talk noises. This results in the channel cross-talk when coupling multiple SOA gate signals. Thus, to quantify the signal impairment introduced by the cross-talk noises, the ON/OFF power ratio of SOA gates is measured under different driving currents as shown in Fig. 4e. It is shown that SOA gates achieve higher than 60 dB ON/OFF ratio under a driving current of 60 mA. The output spectra of SOA gate are also illustrated in Fig. 4e under ON and OFF states, in which the driving current is set to 60 mA at the ON state. The bit error rate (BER) curve of the SOA-based optical switch is also measured under an SOA driving current of 60 mA, as shown in Fig. 4f. Note that the SOA gate can amplify the optical signals and compensate splitting losses of broadcast and select architecture, and no EDFA is applied for the BER measurement. The back-to-back(B2B) BER curve is also recorded as the benchmark. Results indicate that an error-free operation is achieved with a power penalty of 0.5 dB at BER of 1E-9.

## Discussion

We have presented and experimentally demonstrated a nanosecond optical switching and control system for optical DCNs based on the label control mechanism, OFC protocol, and the clock distribution to enable nanosecond data recovery. Optical label channels deliver the allocated time, the label signals for the nanosecond packets forwarding, and the OFC protocol signals to resolve the packet contention. Experimental results confirmed an overall 43.4 ns optical switching and control system operation, 3.1 ns data recovery time without BCDR receivers, and a packet loss rate less than 3.0E-10 after 10 days of continuous and stable network operation. Those results pave the way to the practical deployment of high capacity and low latency optical DCN architectures based on distributed nanosecond optical switches with the nanosecond control system.

High-radix switches in large-scale networks can reduce switch count and hops, thereby decreasing the flow completion time and power consumption. The radix of currently proposed fast optical switches is less than that of the electrical switches, even if the feature of the theoretically unlimited optical bandwidth per port can properly compensate for this radix deficiency. For the next research step, the design of large-radix optical switches will further improve the network performance with the capability to build a fully flat network. Moreover, the flexible adjustment of the optical packet length in real-time is of key importance to adapt the network workload and thereby fully utilize the optical bandwidth. The automatic adjustment mechanism based on the traffic load prediction is a promising solution for future research. Furthermore, a fast and scalable hardware design of Push-In-Extract-Out (PIEO) scheduler proposed and implemented in refs. [28,29] could be deployed in the FPGA-based ToR to flexible manage the optical packet forwarding and thereby improving the network throughput. WDM channels destining to different racks can be deployed at the optical ToR to improve the switching capability. Combining the linear regression algorithm proposed

in ref. [30], the WDM wavelength could be fast tuned to adapt the various traffic pattern.

## Methods

**Traffic model**. The Spirent Ethernet Test Center is configured by the XML file to generate the burst traffic pattern in this experimental demonstration, emulating the real data center traffic characteristics. The Spirent is programmed to generate Ethernet frames with the length varying from 64 bytes to 1518 bytes at the load from 0 to 1. The same as the practical network, 35% of the frames with lengths shorter than 200 bytes are generated as the control frames. More than 45% of the frames with the lengths longer than 1400 bytes are utilized to carry the real application information. Traffic flow is defined as the Spirent-generated continuous Ethernet frames with the same destination within a certain period of time. The flow model is built based on the ON/OFF period length (with/without traffic flow generating), following the data center traffic behavior described in ref. [27].

**Scheduling mechanism**. For the scheduling mechanism at the FPGA-based switch controller in the proposed switching and control system, the controller computes a schedule to guide packet contention resolution and data transmission based on the label request signals, which are delivered on the independent label channels. Note that the OFC protocol and time synchronization are implemented by reusing the label channels, which significantly simplify the scheduling mechanism compared with the conventional schemes. In a conventional implementation, admission control components at ToRs report demand information to the controller and hold data for transmission until triggered to send it by the scheduler. The conventional scheduler in the controller uses a complex scheduling algorithm to determine when to transmit data and how to configure switches. As a comparison, the optical data in the proposed scheduling scheme is not held and transmission is triggered by the scheduler. The optical data packet is directly forwarded with the label signals in the same time slot. The proposed scheduling algorithm then focuses only on the switch configuration to forward the data packets. Benefitting from the parallel processing capabilities of FPGA-based switch controller, the proposed scheduling scheme can be easily scaled out to support more than 64 ToRs in one cluster.

**High data rate**. This proposed optical switching and control system can scale to a higher date rate at NIC in two ways. (1) Adding more transceivers at the data channels. (2) Deploying the optical transceivers with higher speed such as QSFP/SFP28/QSFP-DD or other types at a higher data rate is expected in the future. In this proposed system, the clock is distributed by the label channel to drive the transceivers of the data channels, and this clock can be at lower data rate with respect to the higher data rate transceivers carrying the optical data. The distributed clock can be frequency multiplicated at the CDR block according to the higher data rate of the transceivers. Therefore, the data rate of the NIC for the data channel can be scaled to higher data rate using multiple transceivers or higher speed transceivers.

## Data availability

The data used to produce the plots within this paper are available at https://zenodo.org/record/6371001.

## Code availability

The code used to produce the plots within this paper are available at https://zenodo.org/record/6371001.

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

## Author contributions

X.X. programmed all the FPGA hardware and carried out the experimental investigations; N.C. supervised and directed this project; X.X. and N.C. wrote the manuscript. All authors commented on the manuscript.

## Competing interests

The authors declare no competing interests
