## [Peer Review File · Nature Communications]

Reviewers' comments:

Reviewer #1 (Remarks to the Author):

The authors worked on optical datacenter networks that are able to use optical communication and switching techniques to handle the traffic within the datacenters in a cost- and energy-efficient manner. The major challenge that the paper addresses is nanosecond switching ability. The paper shows very strong experimental demonstration, which reflects great engineering work has been carried out. On the other hand, the technical novelty is a concern, where no clear technology breakthroughs are shown on new optical switching architecture, new switching techniques, or new contention resolution scheme for optical packet switching.

Regarding experimental results, there are also some concerns.

1) The authors reported based on their experiments the overall optical switching and control system can achieve 43.4 ns and the received traffic data is recovered within 3.1 ns. The figures are reported in nanoseconds. However, if it translated to switching capacity, does it equivalent to several tens Mbps for packet switching? It sounds not the same level of switching capacity compared to its electronic counterpart.

2) For the reported packet loss (less than $3.0E-10$ with lower than $1.54E-10$ fluctuation), it should be highly depended on traffic arrival pattern. For instance, the more bursty the traffic is, the higher packet loss the system might suffer. It does not only depend on average arrival rate and how long the system runs. It needs to be better justified.

Reviewer #2 (Remarks to the Author):

This manuscript presents a demonstration of a ns-scale photonic switch architecture through a proof-of-concept experiment in an FPGA-based test bed. Due to the absence of on-switch buffering, it uses a new label control scheme and a custom flow control protocol. The absence of burst-mode receivers is overcome through clock distribution. While there is some novelty in the manuscript, the following aspects would need to be addressed:

- The scale of the demonstration is very small, and more convincing arguments are needed that the proposed architecture will scale. In particular, the scalability of the scheduler (algorithms) and of the clock distribution (jitter, cabling) should be addressed. Also, does the clock need to be distributed at full line rate?

- How does the collision avoidance scheme depend on traffic patterns? The number of NACKs can be a problem for different patterns.

- Does this architecture scale to higher data rates? NICs operate at 50-200Gb/s today and are expected to scale further. What challenges do you expect in this context?

- Does the network have to be time-slotted? What is the impact on latency?

- In Fig. 5: why is the loopback packet loss rate not zero?

- The following recent work is relevant and (most of it) should be cited/discussed:

- o H. Mehrvar et al., "Scalable Photonic Packet Switch Test-bed for Datacenters", OFC 2016
- o T. Gerard et al., "Packet Timescale Wavelength Switching Enabled by Regression Optimisation", IEEE PTL, Apr. 2020
- o V. Shrivastav, "Fast, Scalable, and Programmable Packet Scheduler in Hardware", ACM Sigcomm 2019
- o V. Shrivastav et al., "Shoal: A Network Architecture for Disaggregated Racks", Usenix NSDI 2019
- o A. Forencich et al., "A Dynamically-Reconfigurable Burst-Mode Link Using a Nanosecond Photonic Switch", IEEE JLT 2020
- o A. Forencich et al., "Corundum: An Open-Source 100-Gbps NIC" IEEE FCCM 2020

- Minor:

o Introduction: Refs [11-12] are outdated – I/O bandwidth scaling can be achieved e.g. through 2.5D/3D packaging such as Foveros, Emib (both Intel) resp. CoWoS (TSMC), or co-packaging (e.g. recent work from Rockley, Ayar, IBM, Facebook, Microsoft).

o p.1, col.2, last paragraph: "Despite the promises HELD by ..."

o p.2, col.1: the argument against BCDR should be rewritten; they are expensive because of their limited market size (mostly for PON, but not for data centers today), but there have been demos of fast locking with very low power at adequate data rates (e.g. I. Ozkaya et al., "A 56 Gb/s burst-mode NRZ optical receiver with 6.8 ns power-on and CDR-Lock time for adaptive optical links in 14nm FinFET CMOS," ISSCC 2018.)

Reviewer #3 (Remarks to the Author):

The paper presents an architecture for an optical-switched data-center network. In the envisioned system, top-of-the-rack (ToR) switches are connected using an SOA-based switch for intra-cluster communication and another SOA-based switch for inter-cluster communication. The system relies on two key components: the first is a scheduler that orchestrates the transmissions over the optical switches (one scheduler per switch) and avoid any collision. The scheduler collects all packet requests for the corresponding switches and issues the grants. The scheduler is also responsible for ensuring (possibly using multicast) that at any given time all racks are receiving some data to to ensure that the receivers CDR are always active. The second mechanism at the core of this proposal is to distribute the clock to simplify the task of the CDR (only phase needs to be recovered) and, hence, enable for fast recovery.

The paper is well-written and the authors do a good job at motivating the need for an optical switches within the data center. I was also impressed by the quality of the prototype consisting of both FPGAs

and optical switches. Unfortunately, as I will detail next, I feel that the proposed architecture falls short of expectations in several aspects and, while promising, the work is too premature for being accepted in a top-tier conference such as Nature communication.

High-level comments

=====

Optical switch

The authors advocate for the use of an SOA-based optical switch but they provide very little insights on the trade-offs of such choice. Based on the book by Testa and Pavesi [1], SOA-based switches seem to suffer from serious scalability issues as the number of SOAs needed (i.e., the crosspoint in the crossbar) grows quadratically with the number of ports and, according to the text, their radix is limited to 32 ports. I can give the authors the benefit of the doubt that more recent SOA-based switches (the book is from 2017) can go beyond that but I was hoping that the authors provided some insights on this matter and discuss their scalability targets/assumptions rather than letting the reader speculate. Apart from the overall cost and power, SOAs also introduce noise, which would need to be compensated by higher launch power and/or aggressive use of FEC. Again, sadly the authors seem to ignore all these issues.

Network architecture

The proposed scheme assumes that some NICs are dedicated to the intra-cluster communication while the rest are reserved for inter-cluster communication. This is particularly troublesome as it would impose a static partition of the intra- vs. inter-cluster bandwidth. Given the bursty nature of data-center traffic, this is very hard to determine at design time and it would drastically reduced the flexibility of the network and, ultimately, it could lead to fragmentation and lower utilization. Even with today's over-subscribed networks, a rack can communicate at full rate with any other rack, regardless of its location. Introducing this constraint makes the whole proposal much less appealing. For example, assuming a 32-port optical switch (see above), a network administrator would have to choose whether to prefer give more bandwidth to the 32 racks belonging to the same cluster or to the remaining others (a typical data-center might have more than a 1,000 racks). This would also create a latency asymmetry as within the same cluster packets would only require one hop while for inter-cluster communication, packets need to be relayed by one additional ToR (in most cases).

Apart from the latency issues, routing through an intermediate hop would also reduce the overall network capacity as now some of the ToR uplinks would be used to forward traffic on behalf of other nodes rather than for sending locally-generated traffic. It would have been useful if the authors could quantify the impact on this through simulations (see comments in the Evaluation paragraph below).

Scheduling

The mechanism proposed by the authors is a classic scheduler-based solution where a controller attached to the switch receives all requests and determine a suitable scheduler. While at the high-level the idea seems sound, the paper omits several key details, which makes it hard to really assess its feasibility in practice. First of all, it's not clear whether each ToR NIC would send multiple requests per cycle or only one. In the latter case, the quality of the schedule would be sub-optimal as typical scheduling algorithms (e.g., ISLIP) rely on multiple iterations to improve the quality of the matching. However, if more requests were to be sent concurrently, this would increase the size of the request packets, requiring higher control-plane bandwidth. Also, based on the description provided by the authors, it looks like a scheduling round is required for each packet (or aggregated packet) to send. This seems rather inefficient because it would mean that even if the TOR has to send a burst of packets back-to-back it would have to pay the scheduling latency for each packets within the burst (as opposed to just once). This does not seem to be fundamental and it should be easy to fix but it would have been easier if the authors would have discussed this explicitly.

The paper also seems to lack a proper flow control mechanism. This does not refer to the optical flow control system that the authors use to avoid concurrent transmissions to the same destination but rather the scenario in which the destination rack does not have any more buffer available to host the incoming packets. For example, this can happen if a rack is receiving data destined to the same servers from other different racks (e.g., the server could be a parameter server in a distributed training system), which would lead to queues building up at the rack. In this case, while the optical switches would still be collision-free the destination rack would have to drop packets. Existing systems typically use layer-2 flow control mechanism such as Ethernet PAUSE frame or PFC to avoid dropping packets in this circumstances but it's not clear how simple/difficult it would be to integrate similar mechanisms here (effectively the schedule should guarantee a reverse path for each communication).

Fast CDR

The authors propose to use the switch controller/scheduler to synchronize the rack transceivers such that the CDR is only required to recover the phase. The results presented in Figure 4 are quite impressive. However, after looking at reference [6], I got a bit worried as in that paper, the authors report a similar experiment (from what I can tell) in Figure 4 where they show that even in presence of clock synchronization, the CDR would still take over 40ns. This could just be due to a different setup but I was also wondering if the explanation might (in part) also be that in this submission the authors use a 10 Gbps network (rather than 25 Gbps as in [6]). I can imagine that the lower is the rate, the larger is the UI and, hence, in proportion phase errors are less critical. If this is correct, however, this is rather worrying because it means that as we scale to 50 or 100G SERDES, the approach proposed by the authors might be less effective.

Evaluation

As I mentioned, overall the evaluation is rather solid and it features an impressive setup. Unfortunately, however, the authors only report on relatively simple scenarios, leaving many questions unanswered. For example, as I was mentioning at the beginning, SOA-based switches may introduce significant noise, which ultimately would affect packet errors. While it might be unfeasible to build a 32x32 SOA switch, the authors could emulate the noise introduced by such switch either by adding the corresponding number of SOA-based in a daisy-chain setup or via simulation. This would make the graph in Figure 5 more convincing. Also, it would be useful to report the conventional BER measurements (rather than packet loss) as the latter could be skewed based on the packet size. Further, the authors do not report any measure of end-to-end latency and throughput. Note that the latter can be significantly impacted by the scheduling latency (especially if scheduling occurs on a packet-by-packet basis), by the quality of scheduling (i.e., the ability to always find the correct matching), and also by the 2-hop routing path for inter-cluster communication.. This part might be more suitable for simulation to evaluate the performance at scale. It would be also useful to understand how the computational complexity of the schedule scales with the size of the network. While scheduling over four nodes does not sound super-challenging, scheduling 32 to 64 nodes might significantly increase the 12.4 ns time.

Detailed comments/nits

=====

* "the emerging of latency-sensitive applications are also imposing to DCNs stringent requirement of low latency". This work does not seem to be optimized for latency given the additional overhead due to the scheduling latency and the intermediate relay rack (for inter-cluster communication).

* "However, the implementation of high-bandwidth electrical switches is limited by the application-specific integrated circuits (ASICs) I/O bandwidth due to the scaling issues of the ball grid array (BGA) package¹¹". I'm not sure that this is fully correct as upcoming co-packing optics solutions would enable connecting the ASIC die directly with the optical chipllets on packages without the need of going through the BGA bumps.

* "especial for the intra-DC scenario" -> "especially for ..."

* "the road blocking" -> "the roadblock"

* "controller to all the connected top of racks (ToRs) allowing 3.1 ns data recovery time with no BCDR receivers, and 43.4 ns overall switching and control time of the DCN". The 43.4 ns seems to have a dependency on the size of the network and the 3.1 ns *might* depends on the data rates used (see comments above). Perhaps, it might be fairer to soften this claim

* "while the conflicted packets with lower priority will be forwarded to the ToRs with no destination requesting". Is this step really needed? Few paragraphs below, the authors observe that "the controller [...] exploits the multicast capability of the optical switch to forward packets to the un-destined ToRs to fill the empty slots". If multicast is used, then there is no need for the ToRs that didn't get a grant to send their message to the ToRs with no destination requesting. Am I missing something?

Although we cannot offer to publish your paper in Nature Communications, the work may be appropriate for another journal in the Nature Research portfolio. If you wish to explore suitable journals and transfer your manuscript to a journal of your choice, please use our <https://mts-ncomms.nature.com/cgi-bin/main.plex?el=A7S6Bqrg6A2KpbX5X5A9ftdVR7DS9cvC49TAM1H9MR3gZ> manuscript transfer portal. If you transfer to Nature-branded journals or to the Communications journals, you will not have to re-supply manuscript metadata and files. This link can only be used once and remains active until used.

All Nature Research journals are editorially independent, and the decision to consider your manuscript will be taken by their own editorial staff. For more information, please see our http://www.nature.com/authors/author_resources/transfer_manuscripts.html?WT.mc_id=EMI_NPG_1511_AUTHORTRANSF&WT.ec_id=AUTHOR manuscript transfer FAQ page. Note that any decision to opt in to In Review at the original journal is not sent to the receiving journal on transfer. You can opt in to <https://www.nature.com/nature-research/for-authors/in-review> In Review at receiving journals that support this service by choosing to modify your manuscript on transfer. In Review is available for primary research manuscript types only.

Revised Manuscript and Response to Reviewers' Comments

We would like to thank the reviewers and the editor for their thoughtful comments and helpful suggestions that have significantly improved the manuscript. We have revised the manuscript according to the reviewers' and editors' comments, including the requested explanations as per reviewers' comments and make the originality of our contributions clear.

Responses to the comments are shown in boldface and modifications applied to the revised paper are shown below.

Responses to the comments from reviewer #1

The authors worked on optical datacenter networks that are able to use optical communication and switching techniques to handle the traffic within the datacenters in a cost- and energy-efficient manner. The major challenge that the paper addresses is nanosecond switching ability. The paper shows very strong experimental demonstration, which reflects great engineering work has been carried out. On the other hand, the technical novelty is a concern, where no clear technology breakthroughs are shown on new optical switching architecture, new switching techniques, or new contention resolution scheme for optical packet switching.

We thank the good comments on the strong experimental work from the reviewer.

As summarized also by the Microsoft paper [3], three main technical challenges are preventing the deployment of optical switching technique in data center network: 1) Scalable and nanoseconds control plane; 2) Time synchronization; 3) Burst Clock Data Recovery (CDR). In this work, we propose and experimentally demonstrate a nanoseconds optical switching and control system to comprehensively solve all these challenges. The novel proposed scheme in this paper is based on a combination of a new nanosecond label control and time allocation mechanism for nanoseconds (43.4 ns) switch control and time synchronization, and a precise clock distribution method for nanoseconds (3.1 ns) CDR.

Due to the lack of practical optical buffer, the conflicted packets will cause high packet loss in optically switched network. Besides overcoming the three aforementioned challenges, we also implement the optical flow control (OFC) protocol by reusing the label channels to prevent the packet contention and thus the packet loss.

[3] Ballani, Hitesh, Paolo Costa, Istvan Haller, Krzysztof Jozwik, Kai Shi, Benn Thomsen, and Hugh Williams. "Bridging the last mile for optical switching in data centers." In 2018 Optical Fiber Communications Conference and Exposition (OFC), pp. 1-3. IEEE, 2018.

1) Regarding experimental results, there are also some concerns.

The authors reported based on their experiments the overall optical switching and control system can achieve 43.4 ns and the received traffic data is recovered within 3.1 ns. The figures are reported in nanoseconds. However, if it translated to switching capacity, does it equivalent to several tens Mbps for packet switching? It sounds not the same level of switching capacity compared to its electronic counterpart.

Fig. 1. Bandwidth utilization against inter-packet gap and CDR locking time.

The overall optical control and switch time is 43.4 ns and this includes all the required control (packet contention, flow control, optical switching) as well as the packet guard-time to forward a packet from one ToR to another ToR. This should not be confused with the optical switching only that it is in the order of few ns. Comparing with the electronic switch, switching and control packets from the input port to the output port requires several hundreds of nanoseconds to several microseconds, and also in the case of electronic switch there is a packet guard-time as well. Considering the optical packets in this experiment with a length of 2017 ns (2600 bytes) and the 3.1 ns CDR time, the bandwidth utilization is 97.7% as shown at point (a) in Fig.1. Therefore, the switch capacity mainly depends on the speed of the optical channels. E.g., the switch capacity is 9.77 Gbps for the 10 Gbps optical channel, 39.08 Gbps for the 40 Gbps link, and 97.7 Gbps for the 100 Gbps optical channel.

2) For the reported packet loss (less than $3.0E-10$ with lower than $1.54E-10$ fluctuation), it should be highly depended on traffic arrival pattern. For instance, the more bursty the traffic is, the higher packet loss the system might suffer. It does not only depend on average arrival rate and how long the system runs. It needs to be better justified.

We agree with the reviewer that the packet loss performance also depends on traffic transmitting/arrival pattern, apart from the link rate and system operation time. We apologize that we did not detail the traffic pattern generated in this experiment. In this experiment, the Spirent Ethernet Test Center is configured by the XML file to generate the burst traffic pattern, closing to the

Fig. 2. (a) Cumulative Distribution Function and (b) histogram of the Ethernet frame size; (c) Cumulative Distribution Function of the ON period.

real data center traffic behavior. The Spirent is programmed to generate Ethernet frames with the length varying from 64 bytes to 1518 bytes at the load from 0 to 1. The size Cumulative Distribution Function (CDF) and histogram of the Ethernet frames generated in Spirent are illustrated in Fig. 2(a)(b), following the Bimodal distribution [33-35]. Closing to the practical data center traffic, 35% of the frames with size shorter than 200 bytes are generated as the control frames. More than 45% of the frames with the size longer than 1400 bytes are utilized to carry the real application information.

Traffic flow is defined as the Spirent-generated continuous Ethernet frames with the same destination within a certain period of time. The flow model is built based on the ON/OFF period length (with/without traffic flow generating), following the data center traffic behavior monitored in [34]. The length of traffic flow follows the model of the Pareto Type II distribution. This distribution is characterized by a shape parameter ε , scale parameter θ , and a threshold parameter x_m , which is known as the tail index. The length CDF of the ON period is shown in Equation 1:

$$ONperiod_{CDF}(x) = 1 - \left(1 + \varepsilon \frac{x - x_m}{\theta}\right)^{-\frac{1}{\varepsilon}} \quad x \geq x_m \quad (1)$$

The mean length of the ON period can be calculated by Equation 2:

$$\overline{ONperiod} = x_m + \frac{\theta}{1 - \varepsilon} \quad \varepsilon < 1 \quad (2)$$

In our experiment, we set the ε equaling to 0.9. The θ is 2746, and the x_m is set to 64 bytes in the XML file to configure the Spirent. This guarantees that 80% of the traffic flows are shorter than 10 KB following the flow model reported in [34]. The OFF period is calculated by Equation 3.

$$OFFperiod = ONperiod \left(\frac{1 - load}{load}\right) \quad (3)$$

Fig. 2(c) presents the length CDFs of the ON periods. The traffic flows generated by Spirent on each ON period are randomly destined to one of the possible servers in this experiment. Therefore, the results of packet loss (less than $3.0E-10$ with lower than $1.54E-10$ fluctuation) are validated under the burst traffic pattern (Ethernet frames size and destination, as well as traffic flow length).

[33] T. Benson, A. Akella, and D. A. Maltz, "Network traffic characteristics of data centers in the wild," in *Proceedings of the 10th ACM SIGCOMM conference on Internet measurement*, 2010, pp. 267-280.

[34] T. Benson, A. Anand, A. Akella, and M. Zhang, "Understanding data center traffic characteristics," *ACM SIGCOMM Computer Communication Review*, vol. 40, no. 1, pp. 92-99, 2010.

[35] R. Sinha, C. Papadopoulos, and J. Heidemann, "Internet packet size distributions: Some observations," USC/Information Sciences Institute, Tech. Rep. ISI-TR--643, pp. 1536-1276, 2007.

We have added the text and figures to the Appendix to show the details of the burst traffic model used in our experiment.

In this experimental demonstration, the Spirent Ethernet Test Center is practically configured by the XML file to generate the burst traffic pattern, closing to the real data center traffic behavior. During the measurement, the Spirent is programmed to generate Ethernet frames with the length varying from 64 bytes to 1518 bytes at the load from 0 to 1. The size Cumulative Distribution Function (CDF) and histogram of the Ethernet frames generated in Spirent are illustrated in Fig. 6(a)(b), following the Bimodal distribution [33, 34, 35]. Closing to the practical data center traffic, 35% of the frames with size shorter than 200 bytes are generated as the control frames. More than 45% of the frames with the size longer than 1400 bytes are utilized to carry the real application information.

Traffic flow is defined as the Spirent-generated continuous Ethernet frames with the same destination within a certain period of time. The flow model is built based on the ON/OFF period length (with/without traffic flow generating), following the data center traffic behavior monitored in [34]. The length of traffic flow follows the model of the Pareto Type II distribution. This distribution is characterized by a shape parameter ε , scale parameter θ , and a threshold parameter x_m , which is known as the tail index. The length CDF of the ON period is shown in Equation 1:

$$ONperiod_{CDF}(x) = 1 - \left(1 + \varepsilon \frac{x - x_m}{\theta}\right)^{-\frac{1}{\varepsilon}} \quad x \geq x_m \quad (1)$$

The mean length of the ON period can be calculated by Equation 2:

$$\overline{ONperiod} = x_m + \frac{\theta}{1 - \varepsilon} \quad \varepsilon < 1 \quad (2)$$

In our experiment, we set the ε equaling to 0.9. The θ is 2746, and the x_m is set to 64 bytes in the XML file to configure the Spirent. This guarantees that 80% of the traffic flows are shorter than 10 KB following the flow model reported in [34]. The OFF period is calculated by Equation 3.

$$OFFperiod = ONperiod\left(\frac{1-load}{load}\right) \quad (3)$$

Fig. 6(c) presents the length CDFs of the ON periods. The traffic flows generated by Spirent on each ON period are randomly destined to one of the possible servers in this experiment. Therefore, the results of packet loss (less than 3.0E-10 with lower than 1.54E-10 fluctuation) are validated under the burst traffic pattern (Ethernet frames size and destination, as well as traffic flow length).

Figure 6 | (a) CDF and (b) histogram of the Ethernet frame size; (c) CDF of the ON period.

Responses to the comments from reviewer #2

This manuscript presents a demonstration of a ns-scale photonic switch architecture through a proof-of-concept experiment in an FPGA-based test bed. Due to the absence of on-switch buffering, it uses a new label control scheme and a custom flow control protocol. The absence of burst-mode receivers is overcome through clock distribution. While there is some novelty in the manuscript, the following aspects would need to be addressed:

We thank the good comments on the novelty.

- 1) The scale of the demonstration is very small, and more convincing arguments are needed that the proposed architecture will scale. In particular, the scalability of the scheduler (algorithms) and of the clock distribution (jitter, cabling) should be addressed. Also, does the clock need to be distributed at full line rate?

Figure 2 | Optical DCN based on the nanoseconds optical switching and control system. The total $N \times N$ racks divided into N clusters in the large-scale optical DCN. The two NICs at each ToR can be divided into two categories: NIC of intra cluster (intra-NIC) and NIC of inter cluster (inter-NIC). The $N \times N$ intra-cluster optical switches (IS) connecting the intra-NICs in the same cluster are dedicated for the intra-cluster traffic communication. The inter-NIC of the i -th ToR in different cluster are interconnected by the i -th ($1 \leq i \leq N$) inter-cluster optical switch (ES), which are responsible for the forwarding of inter-cluster traffic.

Figure 2 in the manuscript shows the DCN built based on the proposed optical switch and control system. Note that the intra-cluster interconnect network (consisting of the intra-NICs and intra-cluster switch) and the inter-cluster interconnect network (consisting of the inter-NICs and inter-cluster switch) operate as two independent sub-networks as shown in Fig. 2 by the solid (green color) and broken links (brown color), respectively. Therefore, each sub-network has an independent optical switching and control system with its own label control mechanism, OFC protocol and clock frequency distribution. This is important as the scalability of the optical switching and control system is per cluster scale and not for the whole DCN. This makes the proposed techniques fully distributed and scalable even for DCN with a very large number of ToRs.

For the scheduler at the FPGA-based switch controller in the proposed switch and control system, the controller computes a schedule for packet contention resolution and data transmission based on the label request signals, which are delivered on the independent label channels. Note that the OFC protocol and time synchronization are implemented by reusing the label channels (only one label channel is required between the switch controller and ToR), which significantly simplifies the scheduling mechanism compared with the conventional schemes. In a conventional implementation, admission control components at ToRs report demand information to the controller and hold data for transmission until triggered to send it by the scheduler. The conventional scheduler in the controller uses a complex scheduling algorithm to determine when to transmit data and how to configure switches. As a comparison, the optical data in our scheduling scheme is not held and transmission is triggered by the scheduler. The optical data packet is directly forwarded with the label signals in the same time slot. The scheduling algorithm in the proposed system then focuses only on the switch configuration to forward the data packets. Benefitting from the parallel processing capabilities of FPGA-based switch controller, the proposed scheduling scheme can be easily scaled out to support more than 64 ToRs in one cluster.

Figure 1 | a. Nanoseconds optical switching and control system exploiting label control mechanism, Optical Flow Control protocol and clock distribution mechanism. In the proposed switching control system, N racks are grouped into one cluster and each rack hosts H servers. Each ToR consists of one Ethernet switch and two network interface controllers (NICs). The ToR₁ and switch controller are zoomed to show the detailed functional components. The system consists of the field programming gate array (FPGA)-based ToRs and the SOA-based optical switch with corresponding FPGA-based switch controller. Data channels carrying the optical packets from the ToRs are cross-connected by the optical switch, and the label channels are used to connect the ToRs to the switch controllers.

The continuous clock distribution in the proposed switch and control system is implemented on the independent label channels to each ToR. The distributed clock is recovered and cleaned at the CDR block (equipping a PLL inside) even passing the fiber with different length on label channels, as shown in Fig. 1(a) in the manuscript. The recovered and cleaned clock without jitter is then used to drive the transceivers of the data channels. Therefore, the quality of the continuous distributed clock (jitter and cabling) is not affected by the counts of the label channels (the scalability of the system).

Moreover, the system only needs to distribute the clock frequency to the ToRs in the sub-network, no requirements to align the clock phase as the CDR is done at each ToR in less than one clock cycle (3.1 ns). This is an advantage with respect to other techniques [28], where the phase alignment requires more complex network interconnections and extra devices to guarantee the precise positional relationship between the TX and RX sides, limiting the practical implementation in large scale DCNs.

The clock carried by the label channel do not need to be distributed at full line rate, because the clock can be frequency multiplied at the CDR block according to the transceivers requirements of the data channels.

[28] Clark, K. et al. *Sub-nanosecond clock and data recovery in an optically-switched data centre network. 2018 European Conference on Optical Communication (ECOC), 1-3 (2018).*

We have added the text to the revised manuscript to discuss the scalability of the proposed system.

Note that the intra-cluster interconnect network (consisting of the intra-NICs and intra-cluster switch) and the inter-cluster interconnect network (consisting of the inter-NICs and inter-cluster switch) are two independent sub-networks as shown in Fig. 2 by the solid and broken links, respectively. Each sub-network has an independent optical switching and control system with its own label control mechanism, OFC protocol and clock frequency distribution. This is important as the scalability of the optical switching and control system is per cluster scale and not for the whole DCN. This makes the proposed techniques fully distributed and scalable even for DCN with a very large number of ToRs.

For the scheduler at the FPGA-based switch controller in the proposed switch and control system, the controller computes a schedule for packet contention resolution and data transmission based on the label request signals, which are delivered on the independent label channels. Note that the OFC protocol and time synchronization are implemented by reusing the label channels (only one label channel is required between the switch controller and ToR), which significantly simplify the scheduling mechanism compared with the conventional schemes. In a conventional implementation, admission control components at ToRs report demand information to the controller and hold data for transmission until triggered to send it by the scheduler. The conventional scheduler in the controller uses a complex scheduling algorithm to determine when to transmit data and how to configure switches. As a comparison, the optical data in our scheduling scheme is not hold and transmission triggered by the scheduler. The optical data packet is directly forwarded with the label signals in the same time slot. The scheduling algorithm in the proposed system then focuses only on the switch configuration to forward the data packets. Benefitting from the parallel processing capabilities of FPGA-based switch controller, the proposed scheduling scheme can be easily scaled out to support more than 64 ToRs in one cluster.

The continuous clock distribution in the proposed switch and control system is implemented on the independent label channels to each ToR. The distributed clock is recovered and cleaned at the CDR block (equipping a PLL inside) even passing the fiber with different length on label channels, as shown in Fig. 1(a) in the manuscript. The recovered and cleaned clock without jitter is then used to drive the transceivers of the data channels. Therefore, the quality of the continuous distributed clock (jitter and cabling) is not affected by the counts of the label channels (the scalability of the system). Moreover, the system only needs to distribute the clock frequency to the ToRs in the sub-network, no requirements to align the clock phase as the CDR is done at each ToR in less than one clock cycle (3.1 ns). This is an advantage with respect to other

techniques [28], where the phase alignment requires more complex network interconnections and extra devices to guarantee the precise positional relationship between the TX and RX sides, limiting the practical implementation in large scale DCNs. Furthermore, the clock carried by the label channel do not need to be distributed at full line rate, because the clock can be frequency multiplied at the CDR block according to the transceivers requirements of the data channels.

2) How does the collision avoidance scheme depend on traffic patterns? The number of NACKs can be a problem for different patterns.

Figure 3 | optical packets at the switch output port and optical data packet pattern.

The collision avoidance scheme is the OFC protocol in the manuscript, which does not depend on the traffic patterns but is transparent to them.

In the OFC protocol, the server generated Ethernet frames with different pattern, in terms of types/size, destination and burstiness, are stored at the electrical buffer. Based on the Ethernet MAC address, frames destined to servers in different racks (inter-rack traffic) are stored in the electrical RAM at each network interface controller (NIC). The frames with the same destination are stored in the same buffer block. At each time slot, the copies of the stored frames in the most occupied buffer block are sent to the data packet processor which aggregates the Ethernet frames with the same destination to generate an optical data packet. The pattern of the optical data packet is shown in Fig. 3 in the manuscript. Meanwhile, a corresponding optical label is generated at the label packet processor based on the ToR look-up table to indicate the destination information and forwarding priority of the optical data packet. The optical label (label request packet) is delivered to the switch controller via the label channel. After a small time offset, the corresponding optical data packet is sent out to the optical switch via the data channel.

Therefore, the object of the OFC protocol is the optical data packet, not the Ethernet frames. The Ethernet frames with different traffic pattern is transparency to the OFC protocol (including the number of NACKs) because these frames are aggregated into the optical data packet. The frame patterns just affect the content of the optical data packets.

3) Does this architecture scale to higher data rates? NICs operate at 50-200Gb/s today and are expected to scale further. What challenges do you expect in this context?

Yes, this architecture can scale to higher data rate at NIC in two ways. 1). Adding more transceivers at the data channels. 2) Deploying the optical transceivers with higher speed such as

QSFP/SFP28/QSFP-DD or other types at higher data rate expected in the future. The proposed system supports both cases. In our switch and control system, the clock is distributed by the label channel to drive the transceivers of the data channels, and this clock can be at lower data rate with respect to the higher data rate transceivers carrying the optical data. The distributed clock can be frequency multiplied at the CDR block according to the higher data rate of the transceivers. Therefore, the data rate of the NIC for the data channel can be scaled to higher data rate using multiple transceivers or higher speed transceivers.

We have added the text to the Appendix to discuss the potential challenges for data rate updating.

This proposed system can scale to higher data rate at NIC in two ways. 1). Adding more transceivers at the data channels. 2) Deploying the optical transceivers with higher speed such as QSFP/SFP28/QSFP-DD or other types at higher data rate expected in the future. The proposed system supports both cases. In our switch and control system, the clock is distributed by the label channel to drive the transceivers of the data channels, and this clock can be at lower data rate with respect to the higher data rate transceivers carrying the optical data. The distributed clock can be frequency multiplied at the CDR block according to the higher data rate of the transceivers. Therefore, the data rate of the NIC for the data channel can be scaled to higher data rate using multiple transceivers or higher speed transceivers.

4) *Does the network have to be time-slotted? What is the impact on latency?*

Packet contention could increase the transmission delay because the conflicted packets need to be retransmitted. Deploying the synchronous time-slotted mechanism in the proposed network helps to reduce the probability of optical packet contention and thereby decreasing the extra transmission latency. Moreover, the length of time slot can be flexibly adjusted with the variation of traffic load, which could decrease the impact of buffer queue delay.

Figure 4 | (b) 7 days packet loss rate measured on the optical link. (c) The detailed packet loss rate on the 4th day.

5) *In Fig. 5: why is the loopback packet loss rate not zero?*

The initial state that could introduce a couple of packet loss due to the temporary time disorder. As shown in Fig.4(c) (new results), we report the detailed data of the packet loss rate for the loopback

(ToR1-TX to ToR1-RX) and ToR1 (ToR2-TX, ToR3-TX, ToR4-TX to ToR1-RX) at the fourth day. At the start, there are 2 lost packets at the loopback channel.

We have added the new results and text to explain this question.

The initial state that could introduce a couple of packet loss on the loopback link due to the temporary time disorder. Fig.4(c) shows the detailed data of the packet loss rate for the lookback (ToR1-TX to ToR1-RX) and ToR1 (ToR2-TX, ToR3-TX, ToR4-TX to ToR1-RX), respectively. At the start, there are 2 lost packets at the loopback channel.

6) - *The following recent work is relevant and (most of it) should be cited/discussed:*

H. Mehrvar et al., "Scalable Photonic Packet Switch Test-bed for Datacenters", OFC 2016

T. Gerard et al., "Packet Timescale Wavelength Switching Enabled by Regression Optimisation", IEEE PTL, Apr. 2020

V. Shrivastav, "Fast, Scalable, and Programmable Packet Scheduler in Hardware", ACM Sigcomm 2019

V. Shrivastav et al., "Shoal: A Network Architecture for Disaggregated Racks", Usenix NSDI 2019

A. Forencich et al., "A Dynamically-Reconfigurable Burst-Mode Link Using a Nanosecond Photonic Switch", IEEE JLT 2020

A. Forencich et al., "Corundum: An Open-Source 100-Gbps NIC" IEEE FCCM 2020

The recent work has been cited and discussed in the revised manuscript.

The switch scheduler proposed in [11] needs the extra preamble bits for the clock and data recovery at the start of the wrapped packets, which significantly deteriorated the bandwidth utilization.

WDM channels destining to different racks can be deployed at the optical ToR to improve the switching capability. Combining the linear regression algorithm proposed in [32], the WDM wavelength should be fast tuned to adapt the various traffic pattern.

For the next research step, a fast and scalable hardware design of Push-In-Extract-Out (PIEO) scheduler proposed and implemented in [30][31] could be deployed in the FPGA-based ToR to flexible manage the optical packet forwarding and thereby improving the network throughput.

The burst-mode links in [26] can be dynamically reconfigured to support the fast optical switching. However, the control plane in this solution does not resolve the packet contention, which could cause high packet loss rate.

7) - *Minor:*

1. *Introduction: Refs [11-12] are outdated – I/O bandwidth scaling can be achieved e.g. through 2.5D/3D packaging such as Foveros, Emib (both Intel) resp. CoWoS (TSMC), or co-packaging (e.g. recent work from Rockley, Ayar, IBM, Facebook, Microsoft).*

2. *p.1, col.2, last paragraph: "Despite the promises HELD by ..."*

3. *p.2, col.1: the argument against BCDR should be rewritten; they are expensive because of their limited market size (mostly for PON, but not for data centers today), but there have been demos*

of fast locking with very low power at adequate data rates (e.g. I. Ozkaya et al., "A 56 Gb/s burst-mode NRZ optical receiver with 6.8 ns power-on and CDR-Lock time for adaptive optical links in 14nm FinFET CMOS," ISSCC 2018.)

These errors have been updated in the revised manuscript.

1. Because the pin density can not be increased on the Ball Grid Array (BGA) package [6], the electrical switching technologies are expected to hit the bandwidth bottleneck (>25.6 Tbps) in two generations. New technologies, such as Silicon Photonics [7], 2.5D/3D packaging [8] and co-packaging [9], are being investigated to scale the I/O bandwidth. However, before these technologies becoming viable, a number of challenges have to be solved, e.g., the high complexity to package external laser sources and fiber coupling, the high manufacturing (including both packaging and testing) costs. Moreover, due to the limitations of CMOS scaling, these technologies will ultimately difficult to keep increasing the transistor density.

*[6] H. J. S. Dorren, E. H. M. Wittebol, R. de Kluijver, G. G. de Villota, P. Duan, and O. Raz, "Challenges for Optically Enabled High-Radix Switches for Data Center Networks," *Journal of Lightwave Technology* 33 (2015).*

*[7] D. Thomson et al., "Roadmap on silicon photonics," *Journal of Optics* 18 (2016).*

*[8] England, L., and I. Arsovski. "Advanced packaging saves the day!—How TSV technology will enable continued scaling." In *2017 IEEE International Electron Devices Meeting (IEDM)*, pp. 3-5. IEEE (2017).*

*[9] Janta-Polczynski, Alexander, Elaine Cyr, et al., "Towards co-packaging of photonics and microelectronics in existing manufacturing facilities." In *Optical Interconnects XVIII*, vol. 10538, p. 105380B. International Society for Optics and Photonics (2018).*

2. p.1, col.2, last paragraph: "Despite the promises held by ..."

3. BCDR receivers have been extensively studied in the context of Passive Optical Networks (PONs) and architectures based on gated oscillators or over-sampling have been shown to achieve nanoseconds locking time [24]. These techniques, however, increase the complexity and cost of the transceiver design, and need to be re-evaluated for higher data rates, although initial results are encouraging [25].

*[24] M. Hsieh and G. Sobelman, "Architectures for Multi-Gigabit Wire-Linked Clock and Data Recovery," *IEEE Circuits and Systems Magazine* 8 (2008).*

*[25] A. Rylyakov et al., "A 25 Gb/s Burst-Mode Receiver for Low Latency Photonic Switch Networks," *IEEE Journal of Solid-state Circuits* 50 (2015).*

Responses to the comments from reviewer #3

The paper presents an architecture for an optical-switched data-center network. In the envisioned system, top-of-the-rack (ToR) switches are connected using an SOA-based switch for intra-cluster communication and another SOA-based switch for inter-cluster communication. The system relies on two key components: the first is a scheduler that orchestrates the transmissions over the optical switches (one scheduler per switch) and avoid any collision. The scheduler collects all packet requests for the corresponding switches and issues the grants. The scheduler is also responsible for ensuring (possibly using multicast) that at any given time all racks are receiving some data to to ensure that the

receivers CDR are always active. The second mechanism at the core of this proposal is to distribute the clock to simplify the task of the CDR (only phase needs to be recovered) and, hence, enable for fast recovery.

The paper is well-written and the authors do a good job at motivating the need for an optical switches within the data center. I was also impressed by the quality of the prototype consisting of both FPGAs and optical switches. Unfortunately, as I will detail next, I feel that the proposed architecture falls short of expectations in several aspects and, while promising, the work is too premature for being accepted in a top-tier conference such as Nature communication.:

We thank the good comments and we have added related contents with the reviewer's expectations.

1) Optical switch

The authors advocate for the use of an SOA-based optical switch but they provide very little insights on the trade-offs of such choice. Based on the book by Testa and Pavesi [1], SOA-based switches seem to suffer from serious scalability issues as the number of SOAs needed (i.e., the crosspoint in the crossbar) grows *quadratically* with the number of ports and, according to the text, their radix is limited to 32 ports. I can give the authors the benefit of the doubt that more recent SOA-based switches (the book is from 2017) can go beyond that but I was hoping that the authors provided some insights on this matter and discuss their scalability targets/assumptions rather than letting the reader speculate. Apart from the overall cost and power, SOAs also introduce noise, which would need to be compensated by higher launch power and/or aggressive use of FEC. Again, sadly the authors seem to ignore all these issues.

As the focus of this work is more on the nanoseconds optical switching and control, we have not included an extensive discussion on the SOA-based switch scalability. The choice of the port number of the switch is also very much related to the DCN architecture. We have investigated in the past OPSquare architecture [10] (which is the one used in this work) as well as FOScube architecture [11]. In the OPSquare architecture, the number of ToRs interconnected scales quadratically with SOA-based switch radix. With a switching radix of 64, 4096 ToR and thus 163840 servers (if each rack connects 40 servers) can be interconnected. Considering FOScube architecture for further scaling the interconnected ToRs and thus the servers, with a switching radix of 32, 32768 ToR and thus 1310720 servers (if each rack connects 40 servers) can be interconnected. Therefore, large number of servers can be already interconnected using distributed optical WDM switch with 32 port radix.

A potential issue to scale to large $N \times N$ SOA switch is the noise associated with the gain to compensate the $1 \times N$ broadcast and select. To solve this issue, in OPSquare each cluster is organized in groups [10], each of the WDM transceivers is only dedicated for the communication with a different group of ToRs. For each cluster, the N ToRs are thus divided into p groups and each group contains $F = N/p$ ToRs. One of the p WDM TXs addresses F (instead of N) possible destination ToRs, in combination with the $1 \times F$ optical switch. Thus, the optical radix requirement could decrease from N to F ($F = N/p$). This means that the SOA should compensate less losses of the broadcast and select, leading to less noise and then increase the scalability of the switch.

Fig. 3. OPSquare flat DCN architecture built on fast WDM OXC switches.

[10] Miao, W., Yan, F. and Calabretta, N., 2016. Towards petabit/s all-optical flat data center networks based on WDM optical cross-connect switches with flow control. *Journal of Lightwave Technology*, 34(17), pp.4066-4075.

[11] Yan, F., Xue, X., Pan, B., Guo, X. and Calabretta, N., 2018, September. FOScube: a scalable data center network architecture based on multiple parallel networks and fast optical switches. In *2018 European Conference on Optical Communication (ECOC)* (pp. 1-3). IEEE.

We add the analyzation about the property of SOA based switch in the Appendix to discuss its scalability and noise issues.

The details can be found in the response to the comment-5 (Evaluation)

2) Network architecture

The proposed scheme assumes that some NICs are dedicated to the intra-cluster communication while the rest are reserved for inter-cluster communication. This is particularly troublesome as it would impose a *_static_* partition of the intra- vs. inter-cluster bandwidth. Given the bursty nature of data-center traffic, this is very hard to determine at design time and it would drastically reduced the flexibility of the network and, ultimately, it could lead to fragmentation and lower utilization. Even with today's over-subscribed networks, a rack can communicate at full rate with any other rack, regardless of its location. Introducing this constraint makes the whole proposal much less appealing. For example, assuming a 32-port optical switch (see above), a network administrator would have to choose whether to prefer give more bandwidth to the 32 racks belonging to the same cluster or to the remaining others (a typical data-center might have more than a 1,000 racks). This would also create a latency asymmetry as within the same cluster packets would only require one hop while for inter-cluster communication, packets need to be relayed by one additional ToR (in most cases). Apart from the latency issues, routing through an intermediate hop would also reduce the overall network capacity as now some of the ToR uplinks would be used to forward traffic on behalf of other nodes rather than for sending locally-generated traffic. It would have been useful if the authors could quantify the impact on this through simulations (see comments in the Evaluation paragraph below).

A DC architecture with flexible reconfiguration of the intra-cluster and inter-cluster bandwidth has been investigated and implemented in our published JLT paper [29] with the design of a novel optical

Fig. 4. Schematic of the novel optical ToR. TX: Transmitter; RX: Receiver; WSS: wavelength selective switch; MUX: multiplexer.

ToR as shown in Fig. 4 (the same with the proposed scheme in this manuscript), where the transceivers deployed at the intra-NIC and inter-NIC can be dynamically reassigned under the control of central controller in real-time to adapt the bursty traffic of the intra- and inter-cluster communication. As this paper focuses on the novel nanoseconds optical switching and control, and for a lack of space we cannot report all those architecture details.

We have added a reference [29] and related text in this revised manuscript to explain how to achieve the reconfigurability feature.

The Ethernet switch at each ToR monitors the traffic volume of intra-cluster and inter-cluster communications by reading the destination MAC address [29], and then accordingly assign the adaptable optical bandwidth to the intra-cluster and inter-cluster links.

[29] Xue, X., Yan, F., Prifti, K., Wang, F., Pan, B., Guo, X., Zhang, S. and Calabretta, N., 2020. ROTOS: A Reconfigurable and Cost-Effective Architecture for High-Performance Optical Data Center Networks. *Journal of Lightwave Technology*, 38(13), pp.3485-3494.

For the inter-cluster communication, the Ethernet frames can be sent to the destination by a combination of IS and ES hops. In more details, the traffic of inter-cluster communication is checked (by reading the destination MAC address and comparing the buffer occupation ratio of the intra-cluster and inter-cluster ports) at each intermediate ToR to distinguish if the Ethernet frame belongs to inter-cluster or intra-cluster traffic. The Ethernet switch at the novel optical ToR monitors the buffer occupation ratio of each buffer block in real-time and the detailed scheme can be found in our ECOC-2019 paper [27]. If the buffer of inter-cluster port has lower occupation ratio compared with the intra-cluster port one, the frame will be sent to the inter-cluster link. Otherwise, the traffic will be sent via the intra-cluster link. Therefore, the intermediate ToR is transparent for both the intra-cluster and inter-cluster traffics. Moreover, the mechanism implemented in our work [27] can improve the overall network capacity by selecting the less occupied links (buffer) to forward the inter-cluster traffics.

The latency and capacity performance (see response to the Evaluation paragraph below) has been numerical investigated on the OMNeT++ platform.

[27] Xue, X., Wang, F., Agraz, F., Pagès, A., Yan, F., Pan, B., Spadaro, S. and Calabretta, N., 2019, September. Experimental assessment of SDN-enabled sliceable OPSquare data center network with deterministic QoS. In *45th European Conference on Optical Communication (ECOC 2019)* (pp. 1-4). IET.

3) Scheduling

The mechanism proposed by the authors is a classic scheduler-based solution where a controller attached to the switch receives all requests and determine a suitable scheduler. While at the high-level the idea seems sound, the paper omits several key details, which makes it hard to really assess its feasibility in practice. First of all, it's not clear whether each ToR NIC would send multiple requests per cycle or only one. In the latter case, the quality of the schedule would be sub-optimal as typical scheduling algorithms (e.g., ISLIP) rely on multiple iterations to improve the quality of the matching. However, if more requests were to be sent concurrently, this would increase the size of the request packets, requiring higher control-plane bandwidth.

Each NIC in the optical control system can support multiple requests per cycle. In the proof-of-concept presented in this paper, the NIC only sends one request per cycle because one label request can already guarantee high-quality matching as indicated by the packet loss reported in Fig. 5 (in the original manuscript). Moreover, the length of the label request is 4 bytes, thus the 10 Gb/s label channel has sufficient bandwidth to increase the size of the request packets and also more requests can be sent concurrently.

Also, based on the description provided by the authors, it looks like a scheduling round is required for _each_ packet (or aggregated packet) to send. This seems rather inefficient because it would mean that even if the TOR has to send a burst of packets back-to-back it would have to pay the scheduling latency for each packets within the burst (as opposed to just once). This does not seem to be fundamental and it should be easy to fix but it would have been easier if the authors would have discussed this explicitly.

In the case the ToR has to send a burst of packets back-to-back, which is the intra-rack traffic if we have understood the question correctly, the packets (ethernet frames) will be processed at the Ethernet switch of the ToR and forwarded to the destined servers directly, with no extra scheduling latency.

The paper also seems to lack a proper flow control mechanism. This does not refer to the optical flow control system that the authors use to avoid concurrent transmissions to the same destination but rather the scenario in which the destination rack does not have any more buffer available to host the incoming packets. For example, this can happen if a rack is receiving data destined to the same servers from other different racks (e.g., the server could be a parameter server in a distributed training system), which would lead to queues building up at the rack. In this case, while the optical switches would still be collision-free the destination rack would have to drop packets. Existing systems typically use layer-2 flow control mechanism such as Ethernet PAUSE frame or PFC to avoid dropping packets in this circumstances but it's not clear how simple/difficult it would be to integrate

similar mechanisms here (effectively the schedule should guarantee a reverse path for each communication).

Thank you for this question. Indeed, in the manuscript we did not implement the layer-2 flow control mechanism, like the Ethernet PAUSE frame and priority-based flow control (PFC), but this can be integrated in the control system. The ToR can monitor the buffer occupation ratio of each buffer block in real-time [27]. When the monitored buffer is over the predefined queuing threshold, the ToR can generate a PAUSE or PFC frame and send it back to the corresponding source on the reverse path (label channel) with the normal Ethernet frames.

[27] Xue, X., Wang, F., Agraz, F., Pagès, A., Yan, F., Pan, B., Spadaro, S. and Calabretta, N., 2019, September. Experimental assessment of SDN-enabled sliceable OPSquare data center network with deterministic QoS. In 45th European Conference on Optical Communication (ECOC 2019) (pp. 1-4). IET.

We add the discussion in the revised paper to elaborate this question.

To prevent the packet loss at the overflowed buffer, the layer-2 flow control mechanism, like the Ethernet PAUSE frame and priority-based flow control (PFC), can be integrated in the control system. The ToR can monitor the buffer occupation ratio of each buffer block in real-time. When the monitored buffer is over the predefined queuing threshold, the ToR can generate a PAUSE or PFC frame and send it back to the corresponding source on the reverse path (label channel) with the normal Ethernet frames to pause the traffic transmitting.

4) Fast CDR

The authors propose to use the switch controller/scheduler to synchronize the rack transceivers such that the CDR is only required to recover the phase. The results presented in Figure 4 are quite impressive. However, after looking at reference [6], I got a bit worried as in that paper, the authors report a similar experiment (from what I can tell) in Figure 4 where they show that even in presence of clock synchronization, the CDR would still take over 40ns. This could just be due to a different setup but I was also wondering if the explanation might (in part) also be that in this submission the

Fig. 5. Configurations of IP core GTH (10.3125 Gbps) and GTY (25.6 Gbps).

authors use a 10 Gbps network (rather than 25 Gbps as in [6]). I can imagine that the lower is the rate, the larger is the UI and, hence, in proportion phase errors are less critical. If this is correct, however, this is rather worrying because it means that as we scale to 50 or 100G SERDES, the approach proposed by the authors might be less effective.

As shown in the Fig. 5, the 10 Gbps (10.3125 Gbps) optical links are driven by the IP core GTH in the FPGA, with the typical 32 bits data width. The UI of 10 Gbps is then 3.103 ns (32/10.3125G). The 25 Gbps (25.6 Gbps) optical links are drive by the IP core GTY in the FPGA, with the typical 64 bits data width. The UI of 25 Gbps is then 2.5 ns (64/25.6G), which is almost the same length with the UI of 10 Gbps scenario. The 50 or 100G SERDES could maintain the same UI length with the implementation of high data width IP core.

The better CDR results (3.1ns) achieved in our work, compared with the 40 ns results in ref [6], is mostly because of the short inter-packet gap (IPG) we implemented resulting from the nanoseconds switch control mechanism. As we can see from Fig. 4 in the original manuscript, the shorter IPG could significantly decrease the CDR time. Moreover, The IPG and the idle part of the optical packet are filled with pulse transition sequence ("1010...1010") to maintain the continuous stream of data to the CDR block, similar as in the Ethernet protocol. This mechanism also benefits the 3.1 ns CDR results in our work.

5) Evaluation

As I mentioned, overall the evaluation is rather solid and it features an impressive setup. Unfortunately, however, the authors only report on relatively simple scenarios, leaving many questions unanswered. For example, as I was mentioning at the beginning, SOA-based switches may introduce significant noise, which ultimately would affect packet errors. While it might be unfeasible to build a 32x32 SOA switch, the authors could emulate the noise introduced by such switch either by adding the corresponding number of SOA-based in a daisy-chain setup or via simulation. This would make the graph in Figure 5 more convincing. Also, it would be useful to report the conventional BER measurements (rather than packet loss) as the latter could be skewed based on the packet size. Further, the authors do not report any measure of end-to-end latency and throughput. Note that the latter can be significantly impacted by the scheduling latency (especially if scheduling occurs on a packet-by-packet basis), by the quality of scheduling (i.e., the ability to always find the correct matching), and also by the 2-hop routing path for inter-cluster communication.. This part might be more suitable for simulation to evaluate the performance at scale. It would be also useful to understand how the computational complexity of the schedule scales with the size of the network. While scheduling over four nodes does not sound super-challenging, scheduling 32 to 64 nodes might significantly increase the 12.4 ns time.

In the broadcast and select (B&S) N*N SOA based optical switch, only one SOA optical gate will be ON state when forwarding optical packets to the specific output port. As the reviewer mentioned, other SOA gates at OFF state may not completely block the optical signal and then introduce the cross-talk noises. This results in the channel cross-talk when coupling output signals of SOA-based gates. Thus, to quantify

the signal impairment introduced by the cross-talk noises, the ON/OFF ratio of SOA gates is measured under different driving currents as shown in Fig. 6(a). It is shown that SOA gates achieve higher than 60dB ON/OFF ratio under a driving current of 60 mA. The output spectra of SOA gate are also illustrated in Fig. 6(a) under ON and OFF states, in which the driving current is set to 60mA at the ON state. This demonstrates that when the SOA gate is in the OFF state, the signal is blocked preventing cross-talk noises.

Fig. 6. (a) ON/OFF ratio under different driving currents; (b) BER curve and eye diagram of the SOA switch.

The Bit Error Rate (BER) curve of the SOA based optical switch is measured under an SOA driving current of 60mA, as shown in Fig. 6(b). The Xilinx IBERT IP Core is deployed in each FPGA-based ToR to measure the BER performance of the SOA switch. Note that only the SOA is employed in the optical switch to amplify the optical signals and compensate splitting losses of broadcast & select architecture, and no EDFA is applied for the BER measurement. The back-to-back BER curve is also recorded as the benchmark. Results indicate that an error free operation is achieved with a power penalty of 0.5dB at BER of 1E-9.

The throughput and latency performance as a function of the network scalability has been numerically investigated. An OMNeT++ simulation model of this proposed DCN is implemented with the experimentally measured parameters. The label control mechanism, OFC protocol and clock distribution have been numerically implemented in the OMNeT++ project. Moreover, 4 WDM transceivers are deployed at each ToR and each transceiver equipping with 50 KB buffer and 40 servers are grouped in

Fig. 7. Throughput and server-to-server latency for different network scale.

the same rack. The traffic pattern described in the response to the comment-2 of the reviewer-1 is used in this simulation model. The average link throughput and server end-to-end latency as a function of number of servers are shown in Fig. 7. The throughput baseline is used as a reference. With respect to the baseline, the throughput of network with 40960 servers at the load of 0.5 decreases 8%. The numerical results validate average 10.35% latency degradation as the network scale from 2560 to 40960 servers. The end-to-end latency is below 3 μ s at load of 0.5 for the large scale (40960 servers) network, which indicates the good scalability of the proposed optical DCN.

Figure 2 in the manuscript shows the DCN built based on the proposed optical switch and control system. Note that the intra-cluster interconnect network (consisting of the intra-NICs and intra-cluster switch) and the inter-cluster interconnect network (consisting of the inter-NICs and inter-cluster switch) operate as two independent sub-networks as shown in Fig. 2 by the solid (green color) and broken links (brown color), respectively. Therefore, each sub-network has an independent optical switching and control system with its own scheduling unit. This is important as the scheduling of the optical switching and control system is per cluster scale and not for the whole DCN. Moreover, the cluster size does not impact the scheduling time because the scheduling logic is implemented by the FPGA which features the powerful parallel processing capabilities (32 or 64 bit width per cycle). The scheduling of requests from 32 or 64 ToR is the same 12.4 ns time with the 4 ToRs.

These new evaluation results have been added in the Appendix to discuss the noise and BER performance of SOA-based switch.

In the broadcast and select (B&S) $N \times N$ SOA based optical switch, only one SOA optical gate will be ON state when forwarding optical packets to the specific output port. Other SOA gates at OFF state may not completely block the optical signal and then introduce the cross-talk noises. This results in the channel cross-talk when coupling output signals of SOA-based gates. Thus, to quantify the signal impairment introduced by the cross-talk noises, the ON/OFF ratio of SOA gates is measured under different driving currents as shown in Fig. 7(a). It is shown that SOA gates achieve higher than 60dB ON/OFF ratio under a driving current of 60 mA. The output spectra of SOA gate are also illustrated in Fig. 7(a) under ON and OFF states, in which the driving current is set to 60mA at the

Figure 7 | (a) ON/OFF ratio under different driving currents; (b) BER curve and eye diagram of the SOA switch.

ON state. This demonstrates that when the SOA gate is in the OFF state, the signal is blocked preventing cross-talk noises.

The Bit Error Rate (BER) curve of the SOA based optical switch is also measured under an SOA driving current of 60mA, as shown in Fig. 7(b). The Xilinx IBERT IP Core is deployed in each FPGA-based ToR to measure the BER performance of the SOA switch. Note that only the SOA is employed in the optical switch to amplify the optical signals and compensate splitting losses of broadcast & select architecture, and no EDFA is applied for the BER measurement. The back-to-back BER curve is also recorded as the benchmark. Results indicate that an error free operation is achieved with a power penalty of 0.5dB at BER of 1E-9.

These new evaluation results have been added to investigate the scalability of the proposed system.

Figure 5] Throughput and server-to-server latency for different network scale. The experimentally measured parameters are used to build the OMNeT++ simulation model of this proposed DCN. Moreover, the label control mechanism, OFC protocol, and clock distribution are deployed in the OMNeT++ model, completely following the technical design. 4 WDM transceivers are deployed at each ToR and each transceiver equipping with 50 KB buffer and 40 servers are grouped in the same rack.

The throughput and latency performance as a function of the network scalability has been numerically investigated as well. The experimentally measured parameters are used to build the OMNeT++ simulation model of this proposed DCN. Moreover, the label control mechanism, OFC protocol, and clock distribution are deployed in the OMNeT++ project, completely following the design. 4 WDM transceivers are deployed at each ToR and each transceiver equipping with 50 KB buffer and 40 servers are grouped in the same rack. The traffic pattern described in the Appendix is programmed in this simulation model. The average link throughput and server end-to-end latency as a function of number of servers are shown in Fig. 5. The throughput baseline is used as a reference. With respect to the baseline, the throughput of network with 40960 servers at the load of 0.5 decreases 8%. The numerical results validate average 10.35% latency degradation as the network scale from 2560 to 40960 servers. The end-to-end latency is below 3 μ s at load of 0.5 for the large scale (40960 servers) network, which indicates the fine scalability of the proposed optical DCN.

- 6) * "the emerging of latency-sensitive applications are also imposing to DCNs stringent requirement of low latency". This work does not seem to be optimized for latency given the additional overhead due to the scheduling latency and the intermediate relay rack (for inter-cluster communication).

The discussion on the impact of the scheduling latency can be found in the response to the previous comment. As discussed in the responses to comment-2 (Network architecture), and the numerical assessment reported in Fig. 5, the intermediate rack has limited impact on the latency performance.

- 7) * "However, the implementation of high-bandwidth electrical switches is limited by the application-specific integrated circuits (ASICs) I/O bandwidth due to the scaling issues of the ball grid array (BGA) package¹¹". I'm not sure that this is fully correct as upcoming co-packaging optics solutions would enable connecting the ASIC die directly with the optical chiplets on packages without the need of going through the BGA bumps.

The argument discussing the ASICs I/O bandwidth has been rewritten

Because the pin density can not be increased on the Ball Grid Array (BGA) package [6], the electrical switching technologies are expected to hit the bandwidth bottleneck (>25.6 Tbps) in two generations. New technologies, such as Silicon Photonics [7], 2.5D/3D packaging [8] and co-packaging [9], are being investigated to scale the I/O bandwidth. However, before these technologies becoming viable, a number of challenges have to be solved, e.g., the high complexity to package external laser sources and fiber coupling, the high manufacturing (including both packaging and testing) costs. Moreover, due to the limitations of CMOS scaling, these technologies will ultimately difficult to keep increasing the transistor density.

[6] H. J. S. Dorren, E. H. M. Wittebol, R. de Kluijver, G. G. de Villota, P. Duan, and O. Raz, "Challenges for Optically Enabled High-Radix Switches for Data Center Networks," *Journal of Lightwave Technology* 33 (2015).

[7] D. Thomson et al., "Roadmap on silicon photonics," *Journal of Optics* 18 (2016).

[8] England, L., and I. Arsovski. "Advanced packaging saves the day!—How TSV technology will enable continued scaling." In *2017 IEEE International Electron Devices Meeting (IEDM)*, pp. 3-5. IEEE (2017).

[9] Janta-Polczynski, Alexander, Elaine Cyr, et al., "Towards co-packaging of photonics and microelectronics in existing manufacturing facilities." In *Optical Interconnects XVIII*, vol. 10538, p. 105380B. International Society for Optics and Photonics (2018).

- 8) * "especial for the intra-DC scenario" -> "especially for ..."

* "the road blocking" -> "the roadblock"

These 2 sentences have been rephrased, thanks.

"Especially for the intra-DC scenario where many applications produce short traffic packets."

"Packet contention resolution and fast CDR locking have been the roadblock to the deployment of fast optical switches in the DCNs"

- 9) * "controller to all the connected top of racks (ToRs) allowing 3.1 ns data recovery time with no BCDR receivers, and 43.4 ns overall switching and control time of the DCN". The 43.4 ns seems to have a dependency on the size of the network and the 3.1 ns *might* depends on the data rates used (see comments above). Perhaps, it might be fairer to soften this claim

As discussed in the responses to the Comment-5, the network built based on the proposed optical switch and control system can be divided into two independent sub-networks with its own scheduling unit. Moreover, the switch controller computes a schedule for packet contention resolution and data transmission based on the label request signals, which are delivered on the independent label channels

in each sub-network. Benefitting from the parallel processing capabilities of FPGA-based switch controller, the proposed scheduling scheme can be then scaled out to support more ToRs in one cluster. The scheduling of 32 or 64 ToRs is the same 12.4 ns time with the 4 nodes. Therefore, the 43.4 ns, which consists of 12.4 ns scheduling time, 3 ns switch driver delay, 6 ns switch rising time, 6 ns switch falling time and 16 ns margin time of the switch control signal with respect to the optical packet, is independent to the network size.

The explanation of the 3.1 ns CDR and the data rates can be found in the response to the comment-4.

10) * "while the conflicted packets with lower priority will be forwarded to the ToRs with no destination requesting". Is this step really needed? Few paragraphs below, the authors observe that "the controller [...] exploits the multicast capability of the optical switch to forward packets to the un-destined ToRs to fill the empty slots". If multicast is used, then there is no need for the ToRs that didn't get a grant to send their message to the ToRs with no destination requesting. Am I missing something?

Figure 4 | Data recovery time as the function of the inter-packet gap length for three network cases.

Apologize for the confusion about the multicast. Indeed, thanks to the multicasting option of the switch, any available packet can be sent to the ToRs with no destination requesting. Here in the proposed control system, the choice was to send the conflicted packets to the ToRs with no destination requesting, but other choices are also possible.

We have rephased these sentences to prevent the misunderstanding.

To guarantee this, the switch controller, which has the full vision of the traffic from the ToRs, exploits the multicast capability of the optical switch to forward the conflicted packet with lower priority to one un-destined ToR to fill the empty slot.

REVIEWER COMMENTS

Reviewer #1 (Remarks to the Author):

The authors put great efforts to address the reviewer's comments. Most of them have been solved.

The reviewer still has problems to understand the switching capacity calculation in the authors' reply. Bandwidth utilisation is calculated for a certain line rate. When the line rate is changed, why the bandwidth utilisation is still the same? Besides, the author does not clarify which layer is considered for the switching capacity. The optical packet has 2600 bytes. Somehow is very long for layer 2 switching. The Ethernet frame (including control overhead) ranges from 64 bytes to 1518 bytes. Even for layer 3, the minimal size of the IP packet could be 21 bytes.

Reviewer #2 (Remarks to the Author):

The revised manuscript is improved and addressed my concerns. I have two minor suggestions regarding newly added text:

(1) The following statement is misleading at best: "However, due to the pin density limitation on the Ball Grid Array (BGA) package, the electrical switching technologies are expected to hit the bandwidth bottleneck (>25.6 Tbps) in two generations."

- Do you mean a bandwidth bottleneck being hit at 25.6T or two generations beyond that (102.4T)? In fact, 25.6T switches have been shipping for a while (e.g. Broadcom Tomahawk 4), and there is no fundamental limit to scaling aggregate electrical switching bandwidth from a package to at least 102.4T, and quite possibly well beyond that. Significant industry resources are being spent to keep pushing that limit.

- The pin density of a package is effectively not a limitation, as I/O bandwidth can be intercepted off the top of a package (e.g. co-packaged optics, or high-density electrical IO off the top). A more fundamental limit is the pin density and the maximum bandwidth per pin of the CMOS chips (or chiplets) themselves, as well as the number of chips (or chiplets) that can be used on a single package.

(2) Regarding the statement "Moreover, due to the limitations of CMOS scaling, these technologies will ultimately be difficult to keep increasing the transistor density."

- Please check the grammar.

- While CMOS scaling is slowing, there is a trend towards packaging multiple chiplets onto first-level packages that keep increasing in size; hence the number of transistors per package keeps effectively increasing.

Reviewer #3 (Remarks to the Author):

High-level comments

First of all I want to thank the authors for taking the time and provide a very detailed and thorough rebuttal to all reviewers' comments. That being said, I'm afraid that my original judgment has not changed and I don't think that this manuscript meets the bar in terms of novelty and technical depth for a prestigious journal like Nature Communications.

As the authors pointed out, "the focus of this work is more on the nanosecond optical switching and control" and, hence, this is where I would expect most of the novelty and scientific contribution. At the high level, this boils down to two key components:

1. The design of a scheduler (aka label control mechanism) to arbitrate transmissions and avoid collisions.
2. A fast CDR mechanism to achieve single-digit nanosecond synchronization

The other aspects touched on the papers, e.g., SOA switch and architecture do not seem to be part of the core contribution and already explored in previous publications from the authors.

Unfortunately, neither of the two components highlighted above appear particularly compelling nor novel. The authors provide very limited details on the scheduling design and it's unclear how it differs from traditional iSLIP-like protocols used in many other papers and switch implementation. The authors also do not provide any measurements for scalability or performance for network sizes larger than four nodes. In their rebuttal they argue that "the cluster size does not impact the scheduling time because the scheduling logic is implemented by the FPGA which features the powerful parallel processing capabilities (32 or 64 bit width per cycle)" but it's not clear how the bus width directly relates to the scheduling complexity. For example, the following paper

E. Lago, E. Soto and J. J. Rodriguez-Andina, "Study of FPGA Implementations of Scheduling Algorithms for High-Performance Switches," 2007 IEEE International Symposium on Industrial Electronics, 2007

shows that as you grow your network from a 4x4 to a 16x16 the number of flip-flops (FFs) and lookup tables (LUTs) grow by one or two order of magnitude. For example, for Xilinx FPGAs FFs and LUTs increase by a factor of 18 and 111 respectively. I appreciate that this is a paper from 2007 and FPGA technology have made significant progress and these ratios could be different but I would expect the authors to perform a similar analysis rather than simply claiming that performance is not impacted when moving from 4x4 networks to 64x64 or even higher.

On the second contribution (CDR) front, again, the novelty seems rather limited considering that prior work [Clark et al, 2018] has already observed that by frequency-synchronizing clocks, it is possible to

achieve nanosecond latency. In the rebuttal, the authors argue that they achieve better results by using short inter-packet gaps. While this is an interesting insight, I would not qualify it as a major novel contribution.

Finally, in this new version, the authors also include large-scale simulations to show the scalability of their approach (Figure 5 in the revised manuscript). Although the authors claim that these results "indicate the fine scalability of the proposed optical DCN", I'm not sure I agree with this statement. Looking at the results, it seems that the network can only support up to 50% of the network load, after which the throughput plateaus. This is particularly worrying as one of the key motivations of this work was to overcome bandwidth limitation of existing switches. However, if the result is a network that can only support 50% of the traffic, I'm not sure if the goal is achieved. Of course, there could be multiple reasons for this, e.g., the need to use a 2-hop path for inter-cluster traffic, the inability of the scheduler to find effective matching, or the fact that the traffic pattern is not admissible. Nevertheless, it's up to the authors to clarify this and discuss the reasons and trade-offs involved.

Detailed comments

1. Bandwidth utilization. In the rebuttal, the authors argue that their scheme achieves a 97.7% utilization assuming a 2,600-byte optical packet and 46.5ns (43.4 IPG and 3.1 CDR) control overhead. However, this assumes a 10G transmission. Existing transceivers already operate with 50G SERDESes and 100G SERDESes are expected for the next generation. At this speed, transmitting 2,600-byte packets would only take 208 ns, yielding a bandwidth utilization of less than 80%. In fact, if Ethernet frames (1,500-bytes) were used, the utilization would be even lower (~61%). While it is always possible to increase the optical packet size to compensate for the increase in network speed, this would assume that we have enough aggregate traffic, which may or may not be the case depending on the workload.

2. "one label request can already guarantee high-quality matching as indicated by the packet loss reported in Fig. 5 (in the original manuscript)". Why is packet loss representative of high-quality matching? I would assume that if request cannot be matched, then it will be re-evaluated again at the next iteration, not dropped. A better metric for matching would be to consider the median/99th percentile latency for packets, to indicate how long a packet has to wait before it can be scheduled.

3. "the UI of 10 Gbps is then 3.103 ns (32/10.3125G). The 25 Gbps (25.6 Gbps) optical links are drive by the IP core GTY in the FPGA, with the typical 64 bits data width" I'm not sure I follow the reasoning behind this statement. What I was referring to was the UI in serial clock domain, i.e., before the CDR has taken place. There the UI of 10G signal is 100ps while 25G is 40ps (100G would be 10ps). My argument was that as the (serial) UI length decreases, phase misalignment would lead to more sampling errors and, hence, 3.1ns might not be enough to recover the phase. This has nothing to do with the parallel bus width. However, since the FPGAs used by the authors seem to have the capabilities to run at 25G, I'm wondering why they stuck to 10G rather than providing experiments at higher rate.

4. As already mentioned above, the results in Figure 5 in the revised manuscript are rather concerning as it seems that the system can only support up to 50% of the throughput. This also means that if given

enough time, at higher load the latency will go to infinity as the output rate cannot match the input rate. Therefore, all latency points reported in the chart for load > 0.5 are rather meaningless.

** See Nature Research's author and referees' website at www.nature.com/authors for information about policies, services and author benefits.

Revised Manuscript and Response to Reviewers' Comments

We would like to thank the reviewers and the editor for their thoughtful comments and helpful suggestions that have significantly improved the manuscript. We have revised the manuscript according to the reviewers' and editors' comments, including the requested explanations as per reviewers' comments and make the originality of our contributions clear.

Responses to the comments are shown in boldface and modifications applied to the revised paper are shown below.

Responses to the comments from reviewer #1

The authors put great efforts to address the reviewer's comments. Most of them have been solved.

We thank the good comments on recognizing our efforts and improvements of this manuscript.

- 1) The reviewer still has problems to understand the switching capacity calculation in the authors' reply. Bandwidth utilisation is calculated for a certain line rate. When the line rate is changed, why the bandwidth utilisation is still the same? Besides, the author does not clarify which layer is considered for the switching capacity. The optical packet has 2600 bytes. Somehow is very long for layer 2 switching. The Ethernet frame (including control overhead) ranges from 64 bytes to 1518 bytes. Even for layer 3, the minimal size of the IP packet could be 21 bytes.*

In the bandwidth utilization calculation, the time length of the optical packet is kept equal to 2017 ns for different line rate, e.g., 2600 bytes (2017 ns) for 10 Gb/s and 10400 bytes (2017 ns) for 40 Gb/s line rate. The bandwidth utilization is thus still the same when the line rate is changed.

Figure 1 Optical data packet pattern of 10Gb/s line rate.

Layer 1 (Physical layer) is considered for the switching capacity. As shown in the Fig. 1, the Ethernet frames with the same destination are grouped in an optical packet with length of 2600 bytes for

10 Gb/s links. The optical packet thus can aggregate several Ethernet frames ranging from 64 bytes to 1518 bytes.

Responses to the comments from reviewer #2

The revised manuscript is improved and addressed my concerns.:

We thank the good comments on affirming our efforts.

1) *I have two minor suggestions regarding newly added text:*

(1) The following statement is misleading at best: "However, due to the pin density limitation on the Ball Grid Array (BGA) package, the electrical switching technologies are expected to hit the bandwidth bottleneck (>25.6 Tbps) in two generations."

- Do you mean a bandwidth bottleneck being hit at 25.6T or two generations beyond that (102.4T)? In fact, 25.6T switches have been shipping for a while (e.g. Broadcom Tomahawk 4), and there is no fundamental limit to scaling aggregate electrical switching bandwidth from a package to at least 102.4T, and quite possibly well beyond that. Significant industry resources are being spent to keep pushing that limit.

- The pin density of a package is effectively not a limitation, as I/O bandwidth can be intercepted off the top of a package (e.g. co-packaged optics, or high-density electrical IO off the top). A more fundamental limit is the pin density and the maximum bandwidth per pin of the CMOS chips (or chiplets) themselves, as well as the number of chips (or chiplets) that can be used on a single package.

(2) Regarding the statement "Moreover, due to the limitations of CMOS scaling, these technologies will ultimately be difficult to keep increasing the transistor density."

- Please check the grammar.

- While CMOS scaling is slowing, there is a trend towards packaging multiple chiplets onto first-level packages that keep increasing in size; hence the number of transistors per package keeps effectively increasing.

Many thanks for the reviewer's suggestions. To avoid the misleading, we have updated the text in the revised manuscript. (1). However, due to the limited maximum bandwidth per pin of the CMOS chips, as well as the limited chips number that can be used on a single package, the electrical switching technologies are hard to increase bandwidth linearly. (2). The statement "*Moreover, due to the limitations of CMOS scaling, these technologies will ultimately be difficult to keep increasing the transistor density.*" has been removed in the revised manuscript.

We have updated the text in the revised manuscript to avoid the misleading.

However, due to the limited maximum bandwidth per pin of the CMOS chips, as well as the limited chips number that can be used on a single package, the electrical switching technologies are hard to increase bandwidth linearly. New technologies, such as Silicon Photonics⁷, 2.5D/3D packaging⁸ and co-packaging⁹, are

being investigated to scale the I/O bandwidth. However, before these technologies becoming viable, a number of challenges have to be solved, e.g., the high complexity to package external laser sources and fiber coupling, and the high manufacturing (including both packaging and testing) costs.

Responses to the comments from reviewer #3

First of all I want to thank the authors for taking the time and provide a very detailed and thorough rebuttal to all reviewers' comments.

We thank the good comments and we have added related contents with the reviewer's expectations.

1) That being said, I'm afraid that my original judgment has not changed and I don't think that this manuscript meets the bar in terms of novelty and technical depth for a prestigious journal like Nature Communications.

As the authors pointed out, "the focus of this work is more on the nanosecond optical switching and control" and, hence, this is where I would expect most of the novelty and scientific contribution. At the high level, this boils down to two key components:

1. The design of a scheduler (aka label control mechanism) to arbitrate transmissions and avoid collisions.

2. A fast CDR mechanism to achieve single-digit nanosecond synchronization

The other aspects touched on the papers, e.g., SOA switch and architecture do not seem to be part of the core contribution and already explored in previous publications from the authors.

Unfortunately, neither of the two components highlighted above appear particularly compelling nor novel. The authors provide very limited details on the scheduling design and it's unclear how it differs from traditional iSLIP-like protocols used in many other papers and switch implementation. The authors also do not provide any measurements for scalability or performance for network sizes larger than four nodes. In their rebuttal they argue that "the cluster size does not impact the scheduling time because the scheduling logic is implemented by the FPGA which features the powerful parallel processing capabilities (32 or 64 bit width per cycle)" but it's not clear how the bus width directly relates to the scheduling complexity. For example, the following paper

E. Lago, E. Soto and J. J. Rodriguez-Andina, "Study of FPGA Implementations of Scheduling Algorithms for High-Performance Switches," 2007 IEEE International Symposium on Industrial Electronics, 2007

shows that as you grow your network from a 4x4 to a 16x16 the number of flip-flops (FFs) and lookup tables (LUTs) grow by one or two order of magnitude. For example, for Xilinx FPGAs FFs and LUTs increase by a factor of 18 and 111 respectively. I appreciate that this is a paper from 2007 and FPGA technology have made significant progress and these ratios could be different but I would expect the authors to perform a similar analysis rather than simply claiming that performance is not impacted when moving from 4x4 networks to 64x64 or even higher.

We have summarized in the manuscript four main technical challenges that are preventing the deployment of optical switching technique in data center networks (this has been recognized also by other research center in Microsoft [3]):

- 1) Fast (nanoseconds) switch control mechanism is required to fully utilize the fast optical switches with nanoseconds reconfiguration time;
- 2) Precisely time synchronization. For the control plane to be effective, the ToRs connected to the optical switch must be time-synchronized at a very fine granularity (ideally few nanoseconds) to align the switch control signals and the optical data packets;
- 3) In optically switched network, new physical connections are created every time the switch configuration changes. This implies that the receiver has to continuously recover the clock of the current transmitter to properly sample the incoming signal and recover the data. The longer this process takes, the lower the network throughput will be as no valid data can be received before the Clock Data Recovery (CDR) has been completed;
- 4) One of the main architectural differences between electrical packet switches and optical switches is that the latter are typically bufferless. The lack of optical buffers significantly complicates the design of switch control, as optical data packets must be carefully orchestrated to prevent any collision caused packet loss.

The aforementioned challenges in terms of fast switch control, precise time synchronization, fast CDR locking and packet contention resolution have been the roadblocking to the deployment of fast optical switches in DCNs. To overcome these issues, the networking communities have been investigating various but independent solutions, as most of them focus on solving one or two of those challenges. Moreover, each community address the challenges and problems from their own perspective. Some techniques are promising solutions addressing one certain challenge, but they could also introduce other issues.

In this work, we propose and experimentally demonstrate a nanoseconds optical switching and control system to synergistically solve all these challenges, not only focusing on the scheduling and fast CDR. The novel proposed control system in this paper is based on a combination of a new nanosecond label control and time allocation mechanism for nanoseconds (43.4 ns) switch control and time synchronization, a precise clock distribution method for nanoseconds (3.1 ns) CDR, and the optical flow control (OFC) protocol by reusing the label channels to prevent the packet contention and thus the packet loss.

[3] Ballani, Hitesh, Paolo Costa, Istvan Haller, Krzysztof Jozwik, Kai Shi, Benn Thomsen, and Hugh Williams. "Bridging the last mile for optical switching in data centers." In 2018 Optical Fiber Communications Conference and Exposition (OFC), pp. 1-3. IEEE, 2018.

The label control mechanism (not the scheduler itself) is designed and implemented to arbitrate transmissions, while the packets collisions prevention is based on the developed and implemented Optical Flow Control (OFC) protocol in our manuscript.

In this label control mechanism, Ethernet frames destined to servers in different racks (inter-rack traffic) are stored in the electrical RAM. The copies of the stored frames with the same destination are grouped in an optical data packet. Meanwhile, a corresponding optical label, indicating the destination information and forwarding priority of the optical data packet, is delivered to the switch controller via the label channel. The corresponding optical data packet is sent to the optical switch via the data channel. Based on the received label requests, the switch controller then arbitrates optical packets transmissions.

The developed OFC protocol is deployed on the bidirectional label channels between the switch controller and ToRs. After the contention arbitration, the switch controller sends back to each ToR an ACK signal (indicating packet successfully forwarded by the optical switch) or NACK signal (packet forwarded to the un-destined ToRs). Based on the received ACK/NACK signal, the ToR will release the stored data packet from the electrical RAM (ACK signal) or trigger the data packet processor to retransmit the optical packet (NACK signal). Based on this protocol, the fast switching and control system thus prevents packet loss even with collisions.

The traditional iSLIP-like protocol is a scheduling algorithm that is used in the electrical switches for improving network throughput. The iSLIP-like protocol is not the counterpart with the proposed label control mechanism and OFC protocol that are used to arbitrate transmissions and avoid collisions, respectively, in the optical switching network. The applying object target of iSLIP protocol is the electrical switch, while the label control mechanism and OFC protocol target the whole optical network.

Figure 2 Resource utilization of the switch controller deploying in xc7vx690tffg1761-2.

If we force to carry out the comparison, the data forwarding request in iSLIP-like protocol is firstly sent out from the input port to the output port within the electrical switch. After receiving the request grant signal from the output ports, the data of input ports will be sent to the corresponding

output ports. Thus, the data packets have to wait the grant of forwarding request signals and then to be sent out, significantly increasing the flow completing time. As a comparison, the forwarding request (label) signals and optical data packets in the proposed optical switching and control system are sent out simultaneously in the same timeslot, which effectively decreases the transmission latency.

In the reviewer listed paper,

E. Lago, E. Soto and J. J. Rodriguez-Andina, "Study of FPGA Implementations of Scheduling Algorithms for High-Performance Switches," 2007 IEEE International Symposium on Industrial Electronics, 2007

the scheduling algorithms implemented in this paper are iSLIP and (D)PHM that are used in the electrical switches to improve the network throughput. Again, the results reported in this listed paper cannot be used to evaluate the label control mechanism and OFC protocol that is designed for transmissions arbitration and collisions prevention in the optical switching networks.

Unlike the complicated iSLIP and (D)PHM algorithms, the scheduling logic in the proposed switch controller in our manuscript is quite simple, which is based on the received label request signals to manage the optical packets forwarding and generate the ACK/NACK signals. Following the reviewer's comments, we performed a similar analysis, as shown in Fig. 2, using the method demonstrated in the listed paper. The resource utilization of the overall switch controller is shown in Table 1 for the 3 kinds of network sizes under the different FPGA devices.

Table 1: resource Utilization of the switch controller

Device	Size	FFs (utilization)	LUTs (utilization)
Xilinx xc7vx690tffg1761-2 Low-level device	4x4	27078 (3.13%)	18314 (4.23%)
	16x16	44751 (5.15%)	34536 (7.97%)
	32x32	84248 (9.72)	65982 (15.23)
Xilinx xcvu095-ffvb2104-2-e Intermediate device	4x4	27125 (2.52%)	18456 (3.43%)
	16x16	45086 (4.19%)	35123 (6.53%)
	32x32	84987 (7.90)	66569 (12.38)
Xilinx xcvu9p-flga2104-2L-e High-level device	4x4	27358 (1.16%)	18967 (1.60%)
	16x16	45862 (1.94%)	35479 (3.00%)
	32x32	86136 (3.64)	67413 (5.70)

As we can see from Table. 1, with the network scaling out from a 4x4 to a 32x32, the number of flip-flops (FFs) and lookup tables (LUTs) maintain the same magnitude for the 3 kinds of FPGA devices. Note that the resource utilization reported in Table. 1 is for the overall logic of switch controller, no for the simple scheduling logic. The resource utilization is below 16% even for the low-level device with less resource provisioning, which prove the good scalability of the proposed system.

- 2) *On the second contribution (CDR) front, again, the novelty seems rather limited considering that prior work [Clark et al, 2018] has already observed that by frequency-synchronizing clocks, it is possible to achieve nanosecond latency. In the rebuttal, the authors argue that they achieve better results by using short inter-packet gaps. While this is an interesting insight, I would not qualify it as a major novel contribution.*

The fast CDR achieved in our manuscript is not only because of the short inter-packet gaps (IPG). To further shorten the CDR, the IPG and the idle part of the optical packet are filled with pulse transition sequence ("1010...1010") to maintain the continuous stream of data to the CDR block, similar as in the Ethernet protocol.

Most importantly, the implementation of fast CDR based on clock distribution in our manuscript does not need extra hardware. The clock frequency is distributed by reusing the label channels of the label control mechanism. Moreover, the clock distribution technique, label control mechanism and OFC protocol synergistically operate to solve all the aforementioned 4 issues that limits the deployment of fast optical switches in DCNs.

3) Finally, in this new version, the authors also include large-scale simulations to show the scalability of their approach (Figure 5 in the revised manuscript). Although the authors claim that these results "indicate the fine scalability of the proposed optical DCN", I'm not sure I agree with this statement. Looking at the results, it seems that the network can only support up to 50% of the network load, after which the throughput plateaus. This is particularly worrying as one of the key motivations of this work was to overcome bandwidth limitation of existing switches. However, if the result is a network that can only support 50% of the traffic, I'm not sure if the goal is achieved. Of course, there could be multiple reasons for this, e.g., the need to use a 2-hop path for inter-cluster traffic, the inability of the scheduler to find effective matching, or the fact that the traffic pattern is not admissible. Nevertheless, it's up to the authors to clarify this and discuss the reasons and trade-offs involved.

Figure 3 Throughput for different network scale with deployment of (a) 12.5KB buffer and 4 TRXs (b) 25KB buffer and 6 TRXs.

Figure 5 (Fig. 3 (a) in this rebuttal letter) in the revised manuscript (last version) illustrates and emphasizes the deterioration trend of the network performance as the network scaling from 2560 servers to 40960 servers. Compared with the network deploying 2560 servers, the large network of 40960 servers just has 4.7% and 12.55% throughput degradation even at the high traffic load of 0.5 and 1.0, respectively.

There are 2 reasons for the flat utilization rate after 0.6 load. First, the small size buffer (12.5 KB per transceiver) deployed in the simulation module is faster overflowed at high load. This can cause high packet loss when the load is larger than 0.6. Second, the low oversubscription (4 transceivers at data channel) further deteriorates the buffer overflow caused packet loss and then decrease the bandwidth utilization. We carried out the new numerical investigation with deploying the larger buffer (25 KB per transceiver) and more transceivers (6 transceivers at data channel) in this OMNeT++ module. The results illustrated in Fig. 3 (b) validate the significantly improved throughput performance compared with the results received in the network with 12.5 KB buffer and 4 transceivers.

We have updated the new results and related text in the revised manuscript.

Figure 5] Throughput and server-to-server latency for different network scale. The experimentally measured parameters are used to build the OMNeT++ simulation model of this proposed DCN. Moreover, the label control mechanism, OFC protocol, and clock distribution are deployed in the OMNeT++ model, completely following the technical design. 6 WDM transceivers are deployed at each ToR and each transceiver equipping with 12.5 KB buffer and 40 servers are grouped in the same rack.

The throughput and latency performance as a function of the network scalability has been numerically investigated as well. The average link throughput and server end-to-end latency as a function of number of servers are shown in Fig. 5. The throughput baseline is used as a reference. With respect to the baseline, the throughput of network with 40960 servers at the load of 0.7 decreases 10%. The numerical results validate average 9.46% latency degradation as the network scale from 2560 to 40960 servers. The end-to-end latency is below 3 μ s at load of 0.5 for the large scale (40960 servers) network, which indicates the fine scalability of the proposed optical DCN.

4) Bandwidth utilization. In the rebuttal, the authors argue that their scheme achieves a 97.7% utilization assuming a 2,600-byte optical packet and 46.5ns (43.4 IPG and 3.1 CDR) control overhead. However, this assumes a 10G transmission. Existing transceivers already operate with 50G SERDESes and 100G SERDESes are expected for the next generation. At this speed, transmitting 2,600-byte packets would only take 208 ns, yielding a bandwidth utilization of less than 80%. In fact, if Ethernet frames (1,500-bytes) were used, the utilization would be even lower (~61%). While

it is always possible to increase the optical packet size to compensate for the increase in network speed, this would assume that we have enough aggregate traffic, which may or may not be the case depending on the workload.

The optical packet size increases as the line rate speeding up, and forwarding time of the optical packet is fixed at 2017 ns, e.g., 2600 bytes (2017 ns) for 10 Gb/s and 10400 bytes (2017 ns) for 40 Gb/s line rate. The proposed control system can thus maintain the high bandwidth utilization even for the high line rate. Moreover, due to the optical packet aggregates the Ethernet frames with the same destination, the length of the optical packet must be over the 1500 bytes (the largest frames). For the network deploying the high speed SERDES (50 Gb/s and 100Gb/s), each rack must group more servers, which means sufficient Ethernet frames can be generated to fulfill the longer optical packet. Furthermore, benefitting from the programmable capability of the FPGA-based hardware, the length of the optical data packets can be flexibly reconfigured to adapt the variable traffic load.

5) "one label request can already guarantee high-quality matching as indicated by the packet loss reported in Fig. 5 (in the original manuscript)". Why is packet loss representative of high-quality matching? I would assume that if request cannot be matched, then it will be re-evaluated again at the next iteration, not dropped. A better metric for matching would be to consider the median/99th percentile latency for packets, to indicate how long a packet has to wait before it can be scheduled.

The 'matching' in the manuscript means the precisely alignment of the optical data packets and the corresponding switch control signals, as illustrated in Fig. 4. To align the signals and packets, the ToRs connected to the optical switch must be time-synchronized at a very fine granularity (ideally few nanoseconds). Any inaccuracy in the time synchronization must be compensated by increasing the inter packet gap (IPG), which would reduce the overall bandwidth utilization. Benefitting from the fast label control mechanism and the precise time synchronization mechanism proposed in this manuscript, the switch and control system achieves the super-short 43.4 ns IPG. As we can see from the Fig. 4, the mismatching of the control signals and optical packets will cost packet loss at the switch

Figure 4 Simplified switch control module.

node. That is why we use the packet loss performance as reported in Fig. 5 (in the original manuscript) to evaluate the high-quality matching.

Maybe the matching in the reviewer’s mind should be that the requests are granted or not. Benefitting from the existing of the electrical buffer, in the electrical switches, if request cannot be matched, then it will be re-evaluated again at the next iteration, not dropped. However, due to the lack of the electrical buffer, the conflicted packets with unmatched requests will be dropped and cause packet loss. That is why we developed the OFC protocol to prevent the packet contention caused packet loss in the optical switch and control system.

6) "the UI of 10 Gbps is then 3.103 ns (32/10.3125G). The 25 Gbps (25.6 Gbps) optical links are drive by the IP core GTY in the FPGA, with the typical 64 bits data width" I'm not sure I follow the reasoning behind this statement. What I was referring to was the UI in serial clock domain, i.e., before the CDR has taken place. There the UI of 10G signal is 100ps while 25G is 40ps (100G would be 10ps). My argument was that as the (serial) UI length decreases, phase misalignment would lead to more sampling errors and, hence, 3.1ns might not be enough to recover the phase. This has nothing to do with the parallel bus width. However, since the FPGAs used by the authors seem to have the capabilities to run at 25G, I'm wondering why they stuck to 10G rather than providing experiments at higher rate.

The 10 Gbps links is 32 bits data width and the clock cycle is 3.1 ns. As shown in Fig. 5, the preamble is 32 bits which is sufficient to recover the clock and data. Actually, under most circumstances, the CDR is completed with less than 16 bits. For instance, 12 bits is used to complete the CDR, as the case shown in Fig. 5. Thus, the real CDR time is just 1.16 ns for this case. Due to the random phase position relationship, the CDR time is variable but always less than 32 bits (or 1 clock cycle that is 3.1 ns). Therefore, CDR time is reported as 3.1 ns (but in practice is less than this value).

Figure 5 Transmitted and received packet.

For the 25 Gbps links, the UI in serial clock domain is less than the UI of 10 Gbps links. However, in one clock cycle of 3.1 ns, it can process 78 bits (preamble) for the CDR operation, which is much more than the 32 bits of 10 Gbps links. Even the short UI may have more critical sampling requirement, but with the more bits which means more rising and falling edge for the CDR sampling can accelerate the CDR operation. The FPGA core in our developed board support the 25 Gbps. However, if we want to carry out the experiment of 25 Gbps, we need to buy and update all the hardware, like the 25 Gbps

FMC boards and the transceivers, Spirent Ethernet Test Center. Moreover, the logic (the 25Gbps logic needs the 64 bits width instead of the 32 bits for 10Gbps) and the clock tree (the 25Gbps needs the GTY core, the design of clock tree is totally different from the GTH core for 10Gbps) should be reprogrammed and redesigned due to the links speed up. This is clearly feasible but it is time consuming, we are working to have new personnel to implement this upgrade.

7) *As already mentioned above, the results in Figure 5 in the revised manuscript are rather concerning as it seems that the system can only support up to 50% of the throughput. This also means that if given enough time, at higher load the latency will go to infinity as the output rate cannot match the input rate. Therefore, all latency points reported in the chart for load > 0.5 are rather meaningless.*

The detailed response about the 50% throughput can be found in the response to the question 3. If deploying larger buffer, the high throughput can be supported.

The latency results in Figure 5 in the revised manuscript (last version) are calculated for the packets that complete the transmission. For the situation of the output rate cannot match the input rate, the buffer could be overflowed and cause packet loss. To prevent the packet loss at the overflowed buffer, the layer-2 flow control mechanism, like the Ethernet PAUSE frame and priority-based flow control (PFC), can be integrated in this control system. The ToR can monitor the buffer occupation ratio of each buffer block in real-time. When the monitored buffer is over the predefined queuing threshold, the ToR will generate a PAUSE or PFC frame and send it back to the corresponding source on the reverse path (label channel) with the normal Ethernet frames to pause the traffic transmitting. This avoids data to be stored for long time in the buffer and the latency thus will not go to infinity at the higher load.

REVIEWER COMMENTS

Reviewer #2 (Remarks to the Author):

My comments have been appropriately addressed.

Reviewer #3 (Remarks to the Author):

Before I proceed, I would like to thank the authors for taking the time to carefully respond to my and other reviewers' previous comments. I appreciate how time-consuming this can be and I'm grateful to the authors for their time.

Overall I do appreciate the engineering effort made by the authors to implement the system. However, I'm afraid that I'm still somewhat sceptical on some of the claims made by the authors, which, in my opinion, would deserve further investigation and/or rethinking of the overall architecture.

1) Hardware scalability of the OFC protocol

I acknowledge the effort of the authors to perform an additional analysis of the scalability of their OFC protocol in Table 1 of the revision letter. However, I was somewhat disappointed that they stopped at 32x32 ports. If their goal (as expressed in the introduction) is to provide a solution to the scalability of electrical switches, then I would expect their solution to scale (in terms of number of ports) well beyond the capabilities of today's switches. Existing 25T systems, e.g., Tomhawk 4 can support up to 512 ports operating at 50G and in a 2-3 year time the 102T generation is supposed to scale to 1,024 ports. This is almost two orders of magnitude higher than the scalability figures reported in Table 1. Just to be clear: I was not demanding to implement a 1,000 port switch but in terms of analysis, I would have expected to at least target figures that are comparable (or better) than what state-of-the-art electrical switches can support. Showing results for up to 32 ports seems to somewhat miss the overall goal of this analysis. Also, apart from computation resources, it is also crucial to consider the computation time given the latency-sensitive nature of many network flows (especially for some of the high-bandwidth scenarios like memory disaggregation).

2) Throughput analysis

Again, I thank the authors for having included an additional graph (Figure 3b in the revision letter) seemingly showing that the throughput can scale to much higher value than 50%. However, I found this graph somewhat misleading because looking at the latency is evident that the system cannot sustain a throughput higher than 50% as the latency starts growing very rapidly after such threshold. This is a sign that queues are building up because the system cannot cope with the injected load. Even a tiny mismatch between the injected and offered load will lead to infinite queuing. I suspect that if the experiments were left running for longer, the latency would grow even higher. However, even if just consider the value reports, latency in excess of 10us are hard to tolerate. Today's electrical switches have a latency in the order of ~350ns, i.e., ~22x lower than the values reported for 80% load (~8us).

3) Bandwidth utilization

Thanks for clarifying that the forwarding time of the optical packet size is fixed. This, however, makes me worry because as we scale to 100G and beyond, the packet size would increase significantly. For example, at 100G the L2 packet would be 26,000 bytes, i.e., >17x larger than a Ethernet packet. While the fact that packets are aggregated per rack can certainly help, this would incur additional latency because of batching. Also, at low load, this seems to introduce a 10x higher latency than standard electrical switches (see my comment above), which is rather concerning.

** See Nature Research's author and referees' website at www.nature.com/authors for information about policies, services and author benefits.

Revised Manuscript and Response to Reviewers' Comments

We would like to thank the reviewers and the editor for their thoughtful comments and helpful suggestions that have significantly improved the manuscript. We have revised the manuscript according to the reviewers' and editors' comments, including the requested explanations as per reviewers' comments and make the originality of our contributions clear.

Responses to the comments are shown in boldface and modifications applied to the revised paper are shown below.

Responses to the comments from reviewer #2

My comments have been appropriately addressed.

We thank all the previous comments proposed by the reviewer to improve our manuscript.

Responses to the comments from reviewer #3

Before I proceed, I would like to thank the authors for taking the time to carefully respond to my and other reviewers' previous comments. I appreciate how time-consuming this can be and I'm grateful to the authors for their time. Overall, I do appreciate the engineering effort made by the authors to implement the system. However, I'm afraid that I'm still somewhat sceptical on some of the claims made by the authors, which, in my opinion, would deserve further investigation and/or rethinking of the overall architecture.

First of all, we would like to thank all the previous comments proposed by the reviewer#3 to improve our manuscript, although some of them introducing a few misunderstandings. It's our great honor and definitely our duty to carefully respond these comments.

1) Hardware scalability of the OFC protocol

I acknowledge the effort of the authors to perform an additional analysis of the scalability of their OFC protocol in Table 1 of the revision letter. However, I was somewhat disappointed that they stopped at 32x32 ports. If their goal (as expressed in the introduction) is to provide a solution to the scalability of electrical switches, then I would expect their solution to scale (in terms of number of ports) well beyond the capabilities of today's switches. Existing 25T systems, e.g., Tomhawk 4 can support up to 512 ports operating at 50G and in a 2-3 year time the 102T generation is supposed to scale to 1,024 ports. This is almost two orders of magnitude higher than the scalability figures reported in Table 1. Just to be clear: I was not demanding to implement a 1,000 port switch but in terms of analysis, I would have expected to at least target figures that are comparable (or better) than what state-of-the-art electrical switches can support. Showing results for up to 32 ports seems to somewhat miss the overall goal of this analysis.

Also, apart from computation resources, it is also crucial to consider the computation time given the latency-sensitive nature of many network flows (especially for some of the high-bandwidth scenarios like memory disaggregation).

We would like to clarify that our goal is to provide a scalable optical data center network (DCN) solution, and not to replace 1000 ports electrical switch with 1000 ports optical switch. We have proposed distributed and transparent optical switches with corresponding control to achieve the scalable DCN architecture. As explained in the previous rebuttal letter, the switch with port radix of 32x32 can support a large-scale DCN. We have investigated the DCN scalability as function of the optical switch port radix in the previous OPSquare architecture [1] as well as FOSScube architecture [2]. In the OPSquare architecture, the number of ToRs interconnected scales quadratically with SOA-based switch radix. With a switching radix of 32, 1024 ToR and thus 40960 servers (if each ToR connects 40 servers) can be interconnected. In FOSScube architecture, the number of ToRs interconnected scales cubically, with a switching radix of 32, 32768 (32^3) ToRs and thus 1310720 servers can be interconnected. Moreover, benefitting from the data rate and format transparency, the optical switch can provide much larger bandwidth per port than the one supported by the electrical switch. Thus, without deploying a large number ports, the optical switch can supply the high capacity than the electrical switches. Therefore, large number of servers can be already interconnected using distributed optical switch with 32 port radix. In terms of control, as already explained in previous reviews, the distributed nature of the architecture enables also a distributed control, preventing large computational time as in the case of centralized control. To get the results as shown in Table-1 in the previous response letter, we practically implement the switch controller in the physical FPGA boards. The maximum ports number of the commercial FPGA boards are 32, due to the limitation of GTH/GTY numbers. That is why we did not experimentally analyze the resource utilization of the FPGA-implemented switch controller with more ports. Assuming an FPGA that supports much larger number ports, the results reported in Table-1 will not increase significantly, benefitting from the parallel processing features of FPGA. The resource utilization of scenario with 128x128 ports (xcvu9p-flga2104-2L-e) was roughly estimated with usage of 270000 FFs (11.41% utilization) and 210000 LUTs (17.76% utilization).

[1] Miao, W., Yan, F. and Calabretta, N., 2016. Towards petabit/s all-optical flat data center networks based on WDM optical cross-connect switches with flow control. *Journal of Lightwave Technology*, 34(17), pp.4066-4075.

[2] Yan, F., Xue, X., Pan, B., Guo, X. and Calabretta, N., 2018, September. FOSScube: a scalable data center network architecture based on multiple parallel networks and fast optical switches. In *2018 European Conference on Optical Communication (ECOC)* (pp. 1-3). IEEE.

Table 1: resource utilization and computation time of the switch controller

Device	Size	FFs (utilization)	LUTs (utilization)	Computation Time
Xilinx xc7vx690tffg1761-2 Low-level device	4x4	27078 (3.13%)	18314 (4.23%)	12.4ns (4 clock cycles)
	16x16	44751 (5.15%)	34536 (7.97%)	15.5ns (5 clock cycles)
	32x32	84248 (9.72%)	65982 (15.23%)	15.5ns (5 clock cycles)
Xilinx xcvu095-ffvb2104-2-e	4x4	27125 (2.52%)	18456 (3.43%)	9.3ns (3 clock cycles)
	16x16	45086 (4.19%)	35123 (6.53%)	12.4ns (4 clock cycles)

Intermediate device	32×32	84987 (7.90%)	66569 (12.38%)	12.4ns (4 clock cycles)
Xilinx	4×4	27358 (1.16%)	18967 (1.60%)	6.2ns (2 clock cycles)
xcvu9p-flga2104-2L-e	16×16	45862 (1.94%)	35479 (3.00%)	9.3ns (3 clock cycles)
High-level device	32×32	86136 (3.64%)	67413 (5.70%)	9.3ns (3 clock cycles)

The computation time at the FPGA-based switch controller is shown in Table 1 for the 3 kinds of port radix under the different FPGA devices. The highest computing delay is 15.5 ns (5 clock cycles) for the FPGA device with less resource and processing capability. The 15.5 ns processing time can be compensated by delivering the label request 15.5 ns earlier than the corresponding optical data packet. Thus, the computation time at the switch controller is fast enough to support the time-sensitive applications.

2) Throughput analysis

Again, I thank the authors for having included an additional graph (Figure 3b in the revision letter) seemingly showing that the throughput can scale to much higher value than 50%. However, I found this graph somewhat misleading because looking at the latency is evident that the system cannot sustain a throughput higher than 50% as the latency starts growing very rapidly after such threshold. This is a sign that queues are building up because the system cannot cope with the injected load. Even a tiny mismatch between the injected and offered load will lead to infinite queuing. I suspect that if the experiments were left running for longer, the latency would grow even higher. However, even if just consider the value reports, latency in excess of 10us are hard to tolerate. Today's electrical switches

Figure 1 (a) Throughput and server-to-server latency for different network scale. (b) Latency Cumulative Distribution Function (CDF) of network at different traffic load.

have a latency in the order of ~ 350 ns, i.e., ~ 22 x lower than the values reported for 80% load (~ 8 us).

The latency performance shown in Fig. 1(a) (Fig. 3(b) in the previous revision letter) is the average value, which cannot fully indicate individual differences. Most of the Ethernet frames can be forwarded rapidly without extra queuing, even at the high traffic load. To clearly show the individual latency performance, we programme the OMNeT++ module to calculate the server-to-server delay of

each Ethernet frame, and report the Cumulative Distribution Function (CDF) of server-to-server latency for the network of 2560 servers as illustrated in the new Fig. 1(b). 95% Ethernet frames of high traffic load (0.8) scenario have the same magnitude of latency performance with the low traffic load (0.5) scenario. This is the reason why the throughput can scale to much higher value than 50%, although the average latency starts growing rapidly. Most of the higher average latency at load of 0.8 is contributed by the latency dispersion (less than 5% Ethernet frames) with distribution from 10 μ s to 35.6 μ s. This also explains the average latency of network at 0.8 load is much higher than that of network at 0.5 load.

Please also notice that the latency results of microseconds magnitude reported in Fig. (1) is the server-to-server latency for the overall network and not just for the single switch, consisting of the processing delay of the top of rack (ToR) switch (transmitting server to ToR), processing delay of optical links and switches, and processing delay of the destination ToR (ToR to destination server). The SOA-based optical switch used in the proposed system have a latency in the order of \sim 5ns, much faster than the reported electrical switches.

3) Bandwidth utilization

Thanks for clarifying that the forwarding time of the optical packet size is fixed. This, however, makes me worry because as we scale to 100G and beyond, the packet size would increase significantly. For example, at 100G the L2 packet would be 26,000 bytes, i.e., $>17x$ larger than a Ethernet packet. While the fact that packets are aggregated per rack can certainly help, this would incur additional latency because of batching. Also, at low load, this seems to introduce a $10x$ higher latency than standard electrical switches (see my comment above), which is rather concerning.

Commercial powerful servers with higher capacity generate much more traffic with comparison with the 10G server. Thus, sufficient Ethernet frames can be generated from the powerful servers to fulfill the longer (26,000 bytes) optical packet. Moreover, benefitting from the programmable capability of the FPGA-based hardware, the length of the optical data packets can be flexibly reconfigured to adapt the variable traffic load. As explained in the comment-2, the SOA-based optical switch used in the proposed system have a latency in the order of \sim 5ns, much faster than the reported electrical switches.

REVIEWER COMMENTS

Reviewer #3 (Remarks to the Author):

I would like to thank the authors for having taken the time to go through my comments and for their very detailed responses. This definitely clarified some of the contentious points and I feel that overall they made the contribution much clearer.

I just would like to clarify a few points:

1) My comment re: the switch radix was related to the fact that with today's switches I can interconnect 512 racks (1,024 racks with the 100T switch generation expected in 2-3 years) using a flat hierarchy. Conversely, when using a 32-port switch like in the following proposal, I have to rely on intermediate ToRs to be able to achieve the same scale (1,024 ToRs). This brings two main disadvantages: 10-100x higher end-to-end latency and 50% lower bandwidth in the worst case. On the latency front, the authors are correct that the switch itself only takes 5ns but due to the extra hop and all the corresponding overhead, the end-to-end latency grows up to 10s of ns (compared to a single-hop electrical switch with <500ns latency). This seems to be at odds with the motivation used in the introduction that the system is targeting latency-sensitive applications.

On the bandwidth front, instead, since now each flow takes two hops, in the worst case the bandwidth is reduced by 50%. Another way to see this is that the intermediate destination will have to forward the traffic also on behalf of other nodes, thus reducing the amount of bandwidth for itself. It would be interesting if the authors could provide some quantification of the overall costs/performance trade-offs compared to just using a single switch.

2) Latency: Thanks for adding Figure 1b with the latency CDF. However, this does not really address my previous concern. Even if just a small fraction of packets experience long delays, this is a sign that the system is congested and that queues are building up. The longer you let this run, the higher the tail will be and the average will also be impacted. Again, this is a function of the effect that I was discussing before due to the extra-hop used. If the system was able to support >50% throughput, then there should be no sign of queuing, not even at the tail as it's indeed confirmed by looking at the distribution for load=0.5 in figure.

3) Bandwidth utilization: I agree that recent servers are likely to generate more traffic than previous generations. However, this does not immediately translate into them being able to fill 26KB-packets as this depends on workload-specific factors. For example, many cloud workloads comprise very small RPC requests/responses, which are typically <10KB. In this scenario, due to the hard constraint on packet size, most of the bandwidth would be wasted (possibly more than 50% and in addition to the 50% bandwidth loss discussed above). For example, looking at Facebook traces from the following paper (Figure 5):

* Q. Zhang et al. "High-Resolution Measurement of Data Center Microbursts". IMC 2017

For the cache workload, the most common packet size is <255 bytes. With such workload, the bandwidth utilization with the proposed optical solution would be lower than 1%. Of course, other workloads, e.g., Hadoop can take advantage of larger packets but, again, given the emphasis on latency-sensitive applications (which by their nature use small packets), I found it somewhat counter-intuitive.

**** See Nature Portfolio's author and referees' website at www.nature.com/authors for information about policies, services and author benefits.**

Revised Manuscript and Response to Reviewers' Comments

We would like to thank the reviewers and the editor for their thoughtful comments and helpful suggestions that have significantly improved the manuscript. We have revised the manuscript according to the reviewers' and editors' comments, including the requested explanations as per reviewers' comments and make the originality of our contributions clear. Moreover, some claims have been toned down and a few limitations have also been discussed in the revised manuscript.

Responses to the comments are shown in boldface and modifications applied to the revised paper are shown below.

Responses to the comments from reviewer #3

I would like to thank the authors for having taken the time to go through my comments and for their very detailed responses. This definitely clarified some of the contentious points and I feel that overall they made the contribution much clearer. I just would like to clarify a few points:

First of all, we would like to thank all the previous comments proposed by the reviewer#3 to improve our manuscript, although some of them introducing a few misunderstandings. It's our great honor and definitely our duty to carefully respond these comments.

1)

My comment re: the switch radix was related to the fact that with today's switches I can interconnect 512 racks (1,024 racks with the 100T switch generation expected in 2-3 years) using a flat hierarchy. Conversely, when using a 32-port switch like in the following proposal, I have to rely on intermediate ToRs to be able to achieve the same scale (1,024 ToRs). This brings two main disadvantage: 10-100x higher end-to-end latency and 50% lower bandwidth in the worst case. On the latency front, the authors are correct that the switch itself only take 5ns but due to the extra hop and all the corresponding overhead, the end-to-end latency grows up to 10s of us (compared to a single-hop electrical switch with <500ns latency). This seems to be at odds with the motivation used in the introduction that the system is targeting latency-sensitive applications.

On the bandwidth front, instead, since now each flow takes two hops, in the worst case the bandwidth is reduced by 50%. Another way to see this is that the intermediate destination will have to forward the traffic also on behalf of other nodes, thus reducing the amount of bandwidth for itself. It would be interesting if the authors could provide some quantification of the overall costs/performance trade-offs compared to just use a single switch.

Figure 1. Tomahawk4/BCM56990 interconnection structure.

According to the comments of the last round, Reviewer-3 declared that today's switch Tomahawk4/BCM56990 Series can interconnect 512 racks. However, this switch can only support up to $64 \times 400GbE$, $128 \times 200GbE$, $256 \times 100GbE$, $256 \times 40GbE$, $256 \times 25GbE$, or $256 \times 10GbE$ ports [1]. Selecting the mode of $64 \times 400GbE$, with a reasonable oversubscription ratio of $\frac{2}{5}$ as shown in Fig. 1, 2 optical links from each rack are required to connect the Tomahawk4/BCM56990 switch. This means 2 switch ports of Tomahawk4/ BCM56990 are occupied by each rack. Thus, this switch can only connect $64/2=32$ racks in the mode of $64 \times 400GbE$. Selecting the mode of $128 \times 200GbE$, as shown in Fig.1, each rack needs to occupy 4 switch ports to support a reasonable oversubscription ratio. Thus, this switch can only connect $128/4=32$ racks in the mode of $64 \times 400GbE$. It is impossible to interconnect 512 racks for this Tomahawk4/BCM56990 Series switch.

Figure 2. Next-generation Meta (Facebook) data center network.

Due to the limitations of port radix and bandwidth per port, this switch of Tomahawk4/BCM56990 Series and other equivalent switches cannot build a flat network hierarchy, which still needs intermediate switches to be able to build a large-scale network. In the commercial data center networks [2], as shown in Fig. 2, Tomahawk4/BCM56990 switches are normally deployed as the spine switches, which works as the intermediate switch, to forward the traffic destined different pods. These flows in this electrically switched data center network take three hops (two edge switches and one spine switch) to arrive at the destinations.

Figure 3 | Throughput and server-to-server latency for different network scale. The experimentally measured parameters are used to build the OMNeT++ simulation model of this proposed DCN. Moreover, the label control mechanism, OFC protocol, and clock distribution are deployed in the OMNeT++ model, completely following the technical design. 6 WDM transceivers are deployed at each ToR and each transceiver equipping with 25 KB buffer and 40 servers are grouped in the same rack.

We acknowledge that the intermediate ToRs could worsen the network performance but not as badly as the reviewers commented. Note that the DCN traffic can be divided into intra-rack flows, intra-cluster flows and inter-cluster flows. Most (>50%) of the network traffic is intra-rack flows that are transmitted inside the same rack without passing the optical switches, while the inter-cluster flows have a very low proportion (<12.5%) [3, 4]. In the proposed network, only the inter-cluster flows rely on the intermediate ToRs to arrive at the destinations with 2 hops. Intra-racks traffic and intra-cluster traffic need 0 hop and 1 hop to complete the flow transmission, respectively. Reviewer-3 obviously considers all the traffic flows (100%) taking 2 hops (relying on the intermediate ToRs) to the destinations. That's why Reviewer-3 claimed that in the worst case the bandwidth is reduced by 50% (100% bandwidth/2 hops=50%). In reality, the extra hops caused bandwidth decrease is 6.25% (12.5% bandwidth/2 hops=6.25%) in the worst case. This has been proved by the throughput results as shown in Fig.3 (Fig.5 in the manuscript). The throughput of the network with 40960 servers decreases 7.8% (not 50%) compared with the reference throughput at the load of 0.5.

The end-to-end latency means the server-to-server latency (we have claimed this in the revised manuscript to clearly show this network performance), which consists of the server transmitter processing delay, the ToR+NIC TX processing delay, the optical fiber transmission delay, the optical switch processing delay, the ToR+NIC RX processing delay, and the server receiver processing delay. It is unfair to compare this server-to-server latency with the processing delay (<500ns latency) of a single electrical switch.

We cannot carry out the quantification of the overall costs/performance trade-offs compared to the network just using a single switch, because, as we discussed before, it is impossible to build a fully flat data center network with a single switch to interconnect all the racks, no matter the electrical switches Tomahawk4/BCM56990 or optical switches.

We acknowledge that the port radix of the optical switch is a drawback compared with the electrical switch. We added a new discussion in the revised manuscript to show this limitation.

High-radix switches in large-scale networks can reduce switch count and hops, thereby decreasing the flow completion time and power consumption. The radix of currently proposed fast optical switches is less than that of the electrical switches, even if the feature of the theoretically unlimited optical bandwidth per port can properly compensate for this radix deficiency. For the next research step, the design of large-radix optical switches will further improve the network performance with the capability to build a fully flat network.

The previous claims about the network scalability have been also toned down appropriately to show the deterioration of the network performance caused by the intermediate ToRs.

The throughput and latency performance as a function of the network scalability has been numerically investigated as well. The average link throughput and server server-to-server latency as a function of server number are shown in Fig. 5, where the throughput baseline is used as a reference. With respect to the baseline, the throughput of the network with 40960 servers decreases 10% at the load of 0.7, which is because the intermediate ToRs in the large-scale network have to forward the inter-cluster traffic from other ToRs, thus occupying the bandwidth for its own traffic. The server-to-server latency is below 3 μ s at a load of 0.5 for the large scale (40960 servers) network, and the numerical results validate a limited 9.46% latency degradation as the network scales from 2560 to 40960 servers.

[1] Tomahawk4 / BCM56990 Series: <https://www.broadcom.com/products/ethernet-connectivity/switching/strataxgs/bcm56990-series>

[2] Introducing data center fabric, the next-generation Facebook data center network. <https://engineering.fb.com/2014/11/14/production-engineering/introducing-data-center-fabric-the-next-generation-facebook-data-center-network/>

[3] Benson, Theophilus, Ashok Anand, Aditya Akella, and Ming Zhang. "Understanding data center traffic characteristics." ACM SIGCOMM Computer Communication Review 40, no. 1: 92-99.

[4] Kandula, Srikanth, Sudipta Sengupta, Albert Greenberg, Parveen Patel, and Ronnie Chaiken. "The nature of data center traffic: measurements & analysis." In Proceedings of the 9th ACM SIGCOMM conference on Internet measurement, pp. 202-208.

2)

Latency: Thanks for adding Figure 1b with the latency CDF. However, this does not really address my concern. Even if just a small fraction of packets experience long delays, this is a sign that the system is congested and that queues are building up. The longer you let this run, the higher the tail will be and the average will also be impacted. Again, this is a function of the effect that I was discussing before due to the extra-hop used. If the

system was able to support >50% throughput, then there should be no sign of queuing, not even at the tail as it's indeed confirmed by looking at the distribution for load=0.5 in figure.

Figure 4. Congestion scenario in data center network.

We agree that the long delays of a small fraction of packets due to congestion and queuing will impact the average latency performance. Actually, this is our explanation to answer the comments of reviewer-3 in the last round.

This system congestion and buffer queuing is a natural property for the data center, not a function of the extra-hop effect, no matter electrical or optical switch based network. In an extreme case, as shown in Fig.4, the traffic generated by the 40 servers ($50\text{GbE} \times 40$) in the same rack at a certain time has the same destination, which means all the traffic has to be handled on one single optical link (400GbE). Obviously, this optical link cannot afford this traffic and then congestion occurs. To avoid the packet loss, the congested traffic will be queued in the buffer, which will introduce the extra delay.

3)

*Bandwidth utilization: I agree that recent servers are likely to generate more traffic than previous generation. However, this does not immediately translate into them being able to fill 26KB-packet as this depends on workload specific. For example, many cloud workloads comprises very small RPC requests/responses, which are typically <10KB. In this scenario, due to the **hard** constraint on packet size, most of the bandwidth would be wasted (possibly more than 50% and in addition to the 50% bandwidth loss discussed above). For example, looking at Facebook traces from the following paper (Figure 5):*

** Q. Zhang et al. "High-Resolution Measurement of Data Center Microbursts". IMC 2017*

For the cache workload, the most common packet size is <255 bytes. With such workload, the bandwidth utilization with the proposed optical solution would be lower than 1%. Of course, other workloads, e.g.,

Hadoop can take advantage of larger packets but, again, given the emphasis on latency-sensitive applications (which by their nature use small packets), I found it somewhat counter-intuitive.

Figure 5. Optical data packet pattern.

First of all, the size of the optical packet is 2600 bytes in the manuscript, not the 26KB as the Reviewer-3 considered.

The optical packet aggregates multiple Ethernet frames with various lengths, not only one single Ethernet frame, as shown in Fig.5 (Fig.3 in the manuscript). For the cache workload, the optical packet can aggregate several frames (size <255 bytes) to avoid low bandwidth utilization.

Last but not least, the length of the optical packet is not a fixed and hard constraint. As we explained in the last round, the length of the optical data packets can be flexibly reconfigured to adapt to the variable workload, benefitting from the programmable capability of the FPGA-based hardware.

The really meaningful point is how to automatically and flexibly adjust the length of the optical packets in a network to precisely adapt the workload in real-time. We add a new discussion in the revised manuscript to emphasize the importance of this part of the work in the future.

Note that the length of the optical data packets can be flexibly reconfigured to adapt to the variable workload, benefitting from the programmable capability of the FPGA-based hardware.

Moreover, the flexible adjustment of the optical packet length in real-time is of key importance to adapt the network workload and thereby fully utilize the optical bandwidth. The automatic adjustment mechanism based on the traffic load prediction is a promising solution for future research.